# Genuine quantum scars in many-body spin systems

Andrea Pizzi [1,2] ✉, Long-Hei Kwan[1], Bertrand Evrard [3], Ceren B. Dag[2,4,5] & Johannes Knolle [6,7,8] ✉

Chaos makes isolated systems of many interacting particles quickly thermalize and forget about their past. Here, we show that quantum mechanics hinders chaos in many-body systems: although the quantum eigenstates are thermal and strongly entangled, exponentially many of them are scarred, that is, have an enlarged weight along underlying classical unstable periodic orbits. Scarring makes the system more likely to be found on an orbit it was initialized on, retaining a memory of its past and thus weakly breaking ergodicity, even at long times and despite the system being fully thermal and the eigenstate thermalization hypothesis fulfilled. We demonstrate the ubiquity of quantum scarring in many-body systems by considering a large family of spin models, including some of the most popular ones from condensed matter physics. Our findings, at hand for modern quantum simulators, prove structure in spite of chaos in many-body quantum systems.

Understanding and controlling many-body quantum systems out of equilibrium is a key challenge of modern physics. Left to their isolated dynamics, as becoming increasingly possible in ever improving quantum computers and simulators[1-6], these systems tend to quickly relax to thermal equilibrium, effectively forgetting about their past in the spirit of ergodicity[7-9]. This fate is as universal as tame, and underpinning it is the seemingly chaotic nature of the many-body Hamiltonian. Indeed, the statistical properties of the system's eigenvalues and eigenvectors are in many respect similar to those of certain random matrices[10].

It is known for single-particle systems, however, that chaos can be hindered by quantum scarring[11-13]. This is the phenomenon whereby the quantum wavefunction is enhanced along underlying classical unstable periodic orbits (UPOs). Scarred eigenstates are less random than chaos would suggest, increasing the chances of finding the system on a UPO it was prepared on and challenging the notion of ergodicity[13,14]. Scarring has long been known within single-particle quantum chaos, but its generalization to many-body quantum systems, whose study requires modern quantum simulators and more

advanced numerical tools, has remained virtually unexplored, limited to specific recent instances of interacting bosons in a ring lattice[15] and a periodically driven spin-1 chain[16].

Here, we show that scarring is ubiquitous in many-body systems. For a large family of spin chains, we find that exponentially many eigenstates are enhanced along the UPOs of the associated classical dynamics. Initializing the system on a UPO enhances the probability of finding it on the same UPO at later times. Even in the middle of the spectrum, where entanglement is close to maximal and the eigenstate thermalization hypothesis (ETH) fulfilled[17], we show that the eigenstates are less chaotic than expected and ergodicity is weakly broken.

Our work adds quantum scarring to the (short) list of mechanisms yielding nontrivial effects in many-body quantum systems out of equilibrium, such as integrability[18,19], many-body localization[20-23], Hilbert space fragmentation[24,25], and non-thermal eigenstates in an otherwise chaotic spectrum[26-30]. All these rely on an explicit or emergent partial integrability and host ETH-breaking eigenstates. By contrast, scarring establishes a deviation from chaos in the thermal

[1]Cavendish Laboratory, University of Cambridge, Cambridge, UK. [2]Department of Physics, Harvard University, Cambridge, Massachusetts, USA. [3]Laboratoire Matériaux et Phénoménes Quantiques, Université Paris Cité, Paris, France. [4]Department of Physics, Indiana University, Bloomington, Indiana, USA. [5]ITAMP, Center for Astrophysics, Harvard & Smithsonian, Cambridge, Massachusetts, USA. [6]Department of Physics, Technische Universität München TQM, Garching, Germany. [7]Munich Center for Quantum Science and Technology (MCQST), Munich, Germany. [8]Blackett Laboratory, Imperial College London, London, UK. ✉e-mail: ap2076@cam.ac.uk; j.knolle@tum.de

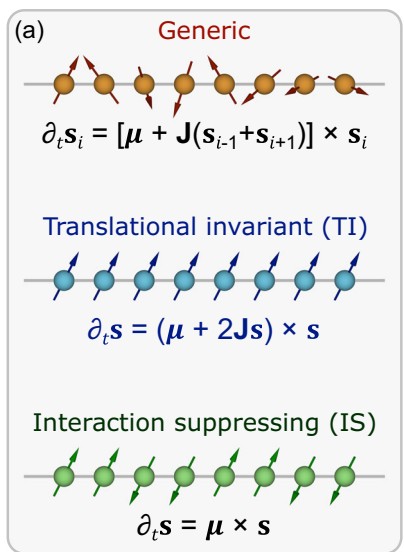

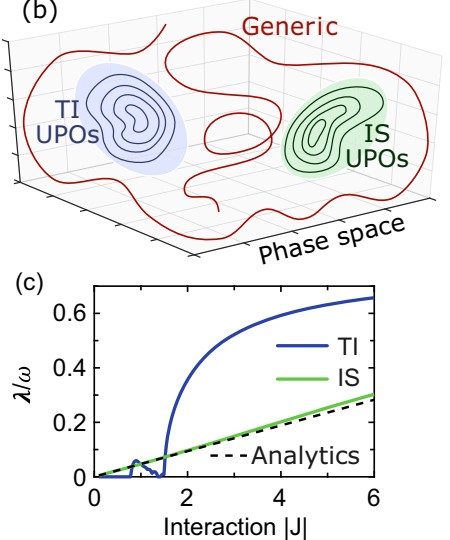

**Fig. 1 | Classical spin dynamics and unstable periodic orbits. a** The dynamics of a classical spin chain is generally chaotic, but yields unstable periodic orbits (UPOs) for the TI states, with aligned spins (blue), and for the IS states, with spins alternating every other site (green). **b** The TI states and IS states form two-dimensional manifolds of UPOs within the highly-dimensional phase space. **c** Quantum scarring is favoured for $\lambda/\omega < 1$, with $\lambda$ the Lyapunov exponent of the UPO and $\omega$ its frequency. This condition holds throughout the whole considered parameter regime, and especially for small $|\mathbf{J}|/|\boldsymbol{\mu}|$. In (**c**) we considered the Ising model with $\boldsymbol{\mu} = (2.4, 0, 0.4)$, $N = 100$, and the UPOs through $\mathbf{s} = \mathbf{y}$. The Lyapunov exponent is computed numerically from the monodromy matrix[59], and analytically (dashed line) for the IS states and $|\mathbf{J}| \ll |\boldsymbol{\mu}|$ (see Supplemental Information).

eigenstates of generic non-integrable many-body systems, where one would least expect it.

A remark on nomenclature is due. Growing attention has been recently devoted to certain many-body quantum systems hosting a few eigenstates which violate ETH and are weakly entangled[27–30]. While these have been dubbed "quantum many-body scars", there is no evidence that they are in fact scars, because they could not be related to UPOs in a chaotic phase space, but rather were often associated to its regular regions[28,31–36]. Such athermal eigenstates are not the focus of our work. Instead, here we consider the thermal eigenstates of many-body systems and show their genuine scarring, due to UPOs in a chaotic phase space.

Beyond[15,16], note that such genuine scarring has also been shown in the Dicke model[37] and for a spinor condensate[19] (also observed in experiments[38]), that with their all-to-all interactions sit somewhere in between few- and many-body systems. Moreover, a semiclassical analysis of the periodic orbits of a quantum many-body system was presented for a periodically driven spin chain in[39], although not in relation to scarring.

## Results

*Model, classical dynamics, and UPOs* − Consider a chain of $N$ spin $s$ particles subject to a magnetic field and nearest-neighbor interactions ($\hbar = 1$)

$$\hat{H} = \sum_{j=1}^{N}\left(\boldsymbol{\mu}\cdot\hat{\mathbf{s}}_j + \frac{1}{s}\hat{\mathbf{s}}_j\mathbf{J}\hat{\mathbf{s}}_{j+1}\right), \tag{1}$$

where $\hat{\mathbf{s}}_j^2 = s(s+1)$, $\hat{\mathbf{s}}_j\mathbf{J}\hat{\mathbf{s}}_{j+1} = \sum_{\alpha,\beta=x,y,z}\hat{s}_j^\alpha J_{\alpha\beta}\hat{s}_{j+1}^\beta$, $J_{\alpha\beta} = J_{\beta\alpha}$, $\boldsymbol{\mu}\cdot\hat{\mathbf{s}}_j = \sum_{\alpha=x,y,z}\mu_\alpha\hat{s}_j^\alpha$, and periodic boundary conditions are assumed. This Hamiltonian is very general, e.g., it describes *any* homogeneous spin 1/2 chain with nearest-neighbor reciprocal interactions, including many prototypical models of condensed matter physics. For instance, the Ising model with both transverse and longitudinal (integrability-breaking) fields is obtained for $\mu_y = 0$ and $\mathbf{J} = J_{zz}\mathbf{z}\otimes\mathbf{z}$, yielding $\hat{H} = \frac{1}{2}\sum_j(\mu_x\hat{\sigma}_j^x + \mu_z\hat{\sigma}_j^z + J_{zz}\hat{\sigma}_j^z\hat{\sigma}_{j+1}^z)$, with $\hat{\sigma}_j^{x,z}$ standard Pauli operators. The Heisenberg, XX, and XXZ models are obtained in a similar way. The

normalization $s^{-1}$ in front of the interaction in Eq. (1) ensures a well-defined limit $s \to \infty$, for which the spins can be described as classical rotors $\{\mathbf{s}_j\}$, with $|\mathbf{s}_j|^2 = 1$ and dynamics[40]

$$\frac{d\mathbf{s}_j}{dt} = \left[\boldsymbol{\mu} + \mathbf{J}\left(\mathbf{s}_{j-1} + \mathbf{s}_{j+1}\right)\right]\times\mathbf{s}_j. \tag{2}$$

The nonlinear dynamics in Eq. (2) is generally chaotic and aperiodic[41]. There are however special families of spin configurations for which the dynamics is periodic instead, see Fig. 1a. One is that of the translationally invariant (TI) states, in which all the spins are aligned, $\{\mathbf{s}_j\} = (\mathbf{s}, \mathbf{s}, \mathbf{s}, \dots)$. Another, for $N$ multiple of 4, is that of the interaction suppressing (IS) states, in which the spins flip at every other site, $\{\mathbf{s}_j\} = (+\mathbf{s}, +\mathbf{s}, -\mathbf{s}, -\mathbf{s}, +\mathbf{s}, +\mathbf{s}, -\mathbf{s}, -\mathbf{s}, \dots)$. These states are special in that the classical dynamics in Eq. (2) does not destroy their nature: a TI state remains such, owing to translational invariance, and a IS state remains such, because $\mathbf{s}_{j-1} + \mathbf{s}_{j+1} = 0$ and the interaction is suppressed. The dynamics from these states is fully specified by the dynamics of just one spin, say $\mathbf{s}$, namely

$$\frac{d\mathbf{s}}{dt} = \begin{cases} (\boldsymbol{\mu} + 2\mathbf{J}\mathbf{s})\times\mathbf{s} & \text{for TI states} \\ \boldsymbol{\mu}\times\mathbf{s} & \text{for IS states} \end{cases}. \tag{3}$$

The vector $\mathbf{s}$ lives on the surface of a sphere, and its Hamiltonian dynamics must be periodic, like any in two dimensions[42].

The many-body dynamics from a TI or IS state will also be periodic but, crucially, generally unstable: a slight perturbation that breaks the TI or IS character of the initial condition leads through chaos to a highly unpredictable and nonperiodic trajectory exploring the many-body phase space ergodically. For the IS states and $|\mathbf{J}| \ll |\boldsymbol{\mu}|$, we compute all the Lyapunov exponents *analytically* (see Supplemental Information), showing that for $N > 4$ the IS states are indeed unstable for all models but trivial ones (e.g., $\lambda = 0$ for the Ising model in a longitudinal field). The TI and IS states thus constitute two manifolds of UPOs within the classical many-body phase space, Fig. 1b. The existence of continuous manifolds of UPOs, rather than isolated UPOs, is a favourable factor for scarring[14].

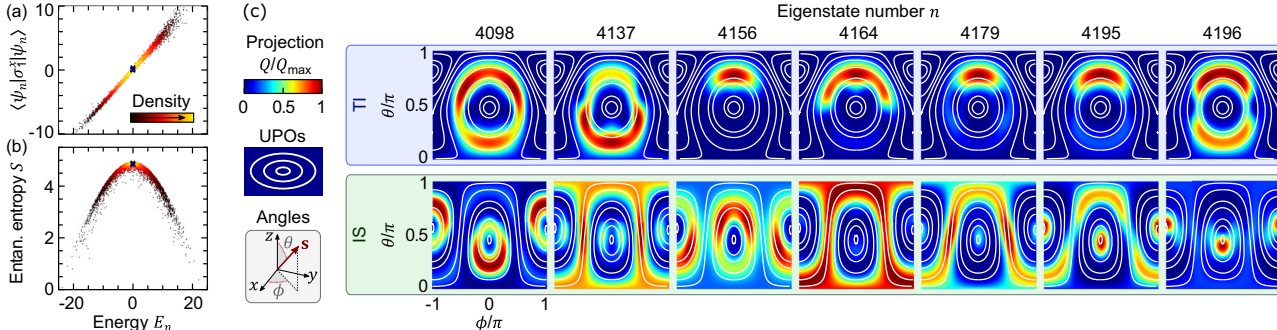

**Fig. 2 | Quantum scars in many-body spin chains. a, b** The many-body eigenstates $|E_n\rangle$ are fully thermal: the expectation value of local observables, e.g., $\langle E_n|\hat{\sigma}_1^x|E_n\rangle$, only depends on the energy[17], as does the bipartite entanglement entropy $S$, which is extensive, $S \sim N$. **c** Projection of selected eigenstates $|E_n\rangle$ onto the TI and IS manifolds of the classical phase space (top and bottom, respectively). The many-body eigenstates are scarred, that is, enhanced along certain UPOs (white lines). The considered eigenstates are marked in (**a, b**) by crosses, and sit in the middle of the thermal spectrum. For each plot we define $Q_{max} = \max_{\theta, \phi} Q(\theta, \phi)$. Here, we considered the Ising model with $s = \frac{1}{2}$, $\boldsymbol{\mu} = (2.4, 0, 0.4)$, $J_{zz} = -1.8$, and $N = 16$.

We note that other such manifolds exist. The IS states are in fact part of a broader manifold of periodic orbits, namely $\{\mathbf{s}_j\} = (\mathbf{s}_1, \mathbf{s}_2, -\mathbf{s}_1, -\mathbf{s}_2, \mathbf{s}_1, \mathbf{s}_2, -\mathbf{s}_1, -\mathbf{s}_2, \dots)$, also yielding $\frac{d\mathbf{s}_i}{dt} = \boldsymbol{\mu} \times \mathbf{s}_i$. We also note that the condition $N = 4k$ led to special effects also in[39], resulting in periodic orbits with enhanced impact on the spectral properties of a periodically driven spin chain, and in[15], relating to "hopping suppressing" UPOs of bosons in a lattice.

As first shown for quantum billiards[13], scarring can be expected when $\lambda/\omega < 1$, with $\lambda$ the Lyapunov exponent of the UPO and $T = 2\pi/\omega$ its period. That is, scars are expected when chaos, acting on a timescale $\sim \lambda^{-1}$, does not prevent a classical trajectory nearby an UPO to return to its neighborhood after one period. In Fig. 1(c) we compute $\lambda/\omega$ for some cases of interest. The IS is unstable in all the considered parameter range, with $\lambda/\omega \sim |\mathbf{J}|/|\boldsymbol{\mu}|$, suggesting that scarring can be enhanced by simply increasing the strength of the field $|\boldsymbol{\mu}|$. The TI state also has $\lambda/\omega < 1$, but becomes stable ($\lambda = 0$) for small $|\mathbf{J}|$. In the following, we consider TI and IS states only when they are unstable ($\lambda > 0$), which is paramount when talking about quantum scars[13].

*Quantum scars* − Let us go back to the quantum many-body problem $\hat{H}|E_n\rangle = E_n|E_n\rangle$. We shall here focus on the deep-quantum limit of $s = \frac{1}{2}$ (larger $s$ are considered in (see Supplemental Information) and yield similar results). The effective size of the Hilbert space is reduced by exploiting the symmetries of the Hamiltonian and of the TI and IS states, i.e., translation by $4n$ sites and mirror reflection, but remains exponentially large in $N$ (for a detailed discussion on symmetries, see Supplemental Information). The eigenstates, attained by exact diagonalization, fulfill the ETH and are characterized by an extensive bipartite entanglement entropy, see Fig. 2a, b. To look for scarring we project the eigenstates onto the classical phase space. Each point $\{\mathbf{s}_i\}$ of the phase space is associated to a product state $|\{\mathbf{s}_i\}\rangle = |\mathbf{s}_1\rangle \otimes |\mathbf{s}_2\rangle \otimes \cdots \otimes |\mathbf{s}_N\rangle$, with $\mathbf{s}_i$ the orientation of the $i$-th quantum spin, $(\mathbf{s}_i \cdot \hat{\mathbf{s}}_i)|\mathbf{s}_i\rangle = s|\mathbf{s}_i\rangle$; the projection of a wavefunction $|\psi\rangle$ on the classical phase space then reads $Q = |\langle\{\mathbf{s}_i\}|\psi\rangle|^2$. Being increasingly accessible in quantum computers and simulators, where for $s = 1/2$ they are often called bitstring probabilities, such projections are quickly emerging as a key object of investigation in many-body quantum chaos[43–50].

The high dimensionality of the classical phase space makes visualizing $Q$ generally complicated. Nonetheless, we are mostly interested in projections along the UPOs, which lie on two-dimensional manifolds parametrized by a polar angle $\theta$ and an azimuth $\phi$. For a few eigenstates in the middle of the spectrum, in Fig. 2c we show the projection $Q$ on the manifolds of TI and IS states. Such projection displays a distinctive feature of scarring, namely it reflects the underlying UPOs and peaks on some. The degree of scarring, as well as which UPOs are responsible for it, varies from eigenstate to eigenstate.

Scarring is particularly remarkable for the IS manifold: not only are all the considered eigenstates in the middle of the spectrum, $E_n \approx 0$, but the whole IS manifold is, because any IS state yields $\langle\{\mathbf{s}_i\}|\hat{H}|\{\mathbf{s}_i\}\rangle = 0$. That is, the structure in $Q$ cannot be due to some UPOs being at a special energy, which makes scarring even more surprising, in analogy with quantum billiards in which the classical orbits are all at the same energy $\frac{p^2}{2m}$[11].

The eigenstates shown in Fig. 2 are selected to showcase the structure of $Q$ in its various shapes and colors. But cherry picking is by no mean required: $Q$ reflects the underlying UPOs for the majority of the eigenstates, which we also show for the XX and XXZ models in (see Supplemental Information). We also note that for the Heisenberg model ($\mathbf{J} = J\mathbb{1}$, not shown), the projection $Q$ of the eigenstates $|E_n\rangle$ is exactly constant along the IS orbits, owing to the underlying $U(1)$ symmetry [$\hat{H}, \sum_j \boldsymbol{\mu} \cdot \hat{\mathbf{s}}_j] = 0$. By contrast, in the models considered here the structure of the eigenstates on the UPOs does not piggyback on any symmetry.

Having seen in Fig. 2 the visual, qualitative features of scarring, we now turn to a quantitative analysis, showing that the eigenstates are anomalously large on the UPOs. We consider the mid-spectrum eigenstates $|E_n\rangle$ of the symmetry sector with zero momentum and left-right mirror parity $+1$, and overlap them with three types of states $|\psi\rangle$: Haar random states, phase-space states $|\{\mathbf{s}_i\}\rangle$, and IS states. To make sure these are treated on par, namely that generic phase-space states are not penalized for being less symmetric than the IS states, we renormalize all the states to have weight 1 on the considered symmetry sector (details in see Supplemental Information). That is, we consider the rescaled projection of $|E_n\rangle$ on $|\psi\rangle$, namely $x = \mathcal{D}\frac{|\langle E_n|\psi\rangle|^2}{\langle\psi|\hat{\mathcal{P}}\hat{\mathcal{P}}^\dagger|\psi\rangle}$, where $\mathcal{D}$ is the size of the considered symmetry sector and $\hat{\mathcal{P}}^\dagger$ the operator projecting on it. Sampling $|E_n\rangle$ and $|\psi\rangle$ yields a probability distribution for $x$, shown in Fig. 3a. Haar random states yield the Porter-Thomas distribution $e^{-x}$[43,51], setting a benchmark for quantum chaotic behaviour. This benchmark is closely followed by the projections of the eigenstates on generic points of the phase space $\{\mathbf{s}_i\}$. A deviation from the benchmark is found for projections on the IS states, yielding a fat tail in the distribution of $x$. This is remarkable: it shows that, due to scarring, the eigenstates can indeed have an anomalously large projection on the UPOs.

To quantify at the same time both aspects of scarring, namely the visual features of Fig. 2 and the fat tail of the projection in Fig. 3a, we introduce a "scarness" parameter $S = 4\mathcal{D} \times \max_{IS} \oint Q_\psi$, where $\oint Q$ denotes averaging of $Q$ along each UPO, and $\max_{IS}$ maximization over the IS UPOs (details in see Supplemental Information). Sampling $|\psi\rangle$ yields a probability distribution for $S$, shown in Fig. 3b. While the

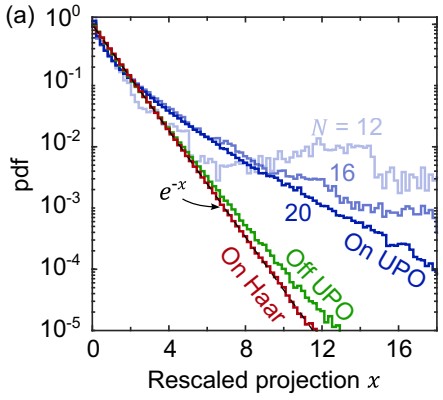
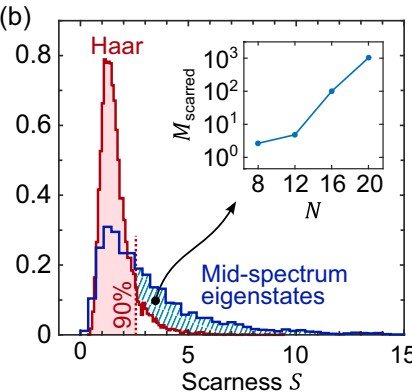

**Fig. 3 | Quantitative features of scarring. a** Distribution of the rescaled overlaps $x$ between mid-spectrum eigenstates and various families of states. The overlaps with Haar random states (red) have distribution $e^{-x}$ (dashed), similarly to the overlaps with generic points $\{s_i\}$ of the phase space (green). By striking contrast, the overlaps with the IS states (blue) yield a much fatter tail, that is, scarring makes some eigenstates anomalously large on the UPOs. **b** The distribution of the scarness parameter $S$ for mid-spectrum eigenstates (blue) has a long tail compared to Haar random states (red). Indeed, the number $M_{\text{scarred}}$ of scarred eigenstates, defined as the number of eigenstates with $S$ larger than the 90-th percentile of the Haar states (red dashed line), minus 10% (see Supplemental Information), is large and appears to grow exponentially with system size (inset). In (**a**, **b**) the eigenstates are uniformly sampled from the middle of the spectrum, namely among the 10% eigenstates with lowest $|E|$. In (**a**), the phase-space states are obtained sampling the spins $\{s_i\}$ uniformly and independently from the sphere, and the IS states are obtained sampling $\mathbf{s}_1$ uniformly from the sphere and alternating the other spins at every other site as in Fig. 1(a). Here, $N = 20$ except where otherwise and explicitly specified. All other parameters are as in Fig. 2.

mid-spectrum eigenstates are often compared to Haar random states, they yield a distribution of the scarness $S$ with a much fatter tail. Indeed, the $S$ of many eigenstates is larger than the $S$ of most Haar states, and the number of scarred eigenstates $M_{\text{scarred}}$ grows exponentially with the considered system sizes $N$ (inset).

The eigenstates are not directly measurable, but the dynamics is, and this must reflect the properties of the eigenstates. Indeed, while the thermal spectrum underpins the equilibration of local observables within short times, the scarring of exponentially many eigenstates leaves a mark on the long-time dynamics of the return probabilities. In analogy to single-particle quantum chaos[13], preparing the system on a UPO increases the probability of finding it on the same UPO at later times. This effect is shown in Fig. 4 by considering two initial conditions: the IS state with axis $\mathbf{y}$, namely $|\psi_0\rangle = \otimes_{i=1}^{N}|\nu_i \mathbf{y}\rangle$, and the IS state with axis $\boldsymbol{\mu}$, namely $|\psi_0\rangle = \otimes_{i=1}^{N}|\nu_i \boldsymbol{\mu}/|\boldsymbol{\mu}|\rangle$, with $\{\nu_i\} = (+ + - - + + - - \ldots)$. For both we numerically integrate the Schrödinger dynamics and compute the time-averaged projection $\bar{Q} = \lim_{t\to\infty}\frac{1}{t}\int_0^t d\tau\, Q(|\psi(\tau)\rangle)$ on the manifold of IS states. For the Ising, XX, and XXZ models in a field, we observe that the system is more likely to be found on the UPO it started from, even long after thermalization.

We emphasize that this effect is not due to the initial condition overlapping with a few non-thermal eigenstates[27], but to many of the thermal eigenstates being scarred. This is a genuinely quantum effect: due to ergodicity, a classical ensemble prepared nearby an IS UPO will at long times spread uniformly across the phase space at $E = 0$, in a way that does not depend on the specific initial condition, see Fig. 4a. By striking contrast, it is more likely to find the quantum system on the UPO it started from. In other words, scarring makes the quantum system remember its past better than a classical system would, in a rare example of weak ergodicity breaking.

We use the adjective *weak* because, while the enhancement of the phase-space projection on the UPOs is large in relative terms, it is small in absolute terms, e.g., $Q_{\text{max}} \sim 5\times 10^{-4}$ in Fig. 2 and $\bar{Q}_{\text{max}} \sim 3\times 10^{-5}$ in Fig. 4. A scaling analysis is presented in Fig. 4b, c, showing that $\bar{Q}$ decays exponentially with $N$ while its relative enhancement does not, exhibiting a memory effect. Due to translation symmetry, the Hilbert space is divided in $N$ momentum sectors, each of size $\sim e^{\mathcal{O}(N)}/N$, and we expect the average overlap to scale as the inverse sector size,

$Q \sim Ne^{-\mathcal{O}(N)}$. Scarring should be seen as a small albeit measurable correction to the picture of thermalization in isolated quantum systems[7,9], not contradicting paradigms such as the ETH[17]. Indeed, as long noted by Srednicki[52], in many-body systems the effect of scarring is mostly washed away when integrating over the phase space, as effectively done when computing the expectation value of simple observables (e.g., $\langle s_j^z \rangle$).

## Discussion

Analysing a broad family of spin chains, including any uniform spin-1/2 chain with nearest-neighbor interactions, our work proves the ubiquity of quantum scarring in many-body systems. In single-particle systems, chaos–and consequently scarring–become meaningful only in the semiclassical regime, such as at sufficiently high energies in quantum billiards[11]. By contrast, many-body systems can achieve a quantum chaotic regime through sufficiently large system sizes $N$. This distinction enables a fundamentally new type of scarring unique to many-body systems. For instance, we found that the $s \to \infty$ classical dynamics in Eq. (2) scars the quantum system all the way down to the deep-quantum limit of $s = \frac{1}{2}$. Scarring enhances the eigenstates $|E_n\rangle$ along certain UPOs and makes a system better remember its past, curbing chaos even in fully thermal and non-integrable many-body systems.

Our predictions can be straightforwardly verified in state-of-the-art quantum simulators for spin Hamiltonians[2–6], opening new possibilities for the experimental observation of scars[53–57]. In particular, by repeatedly preparing a product state, letting it evolve, and measuring it in a different basis, one should be able to show that the system is more likely to be found on the UPO it was prepared on (Fig. 4), which is a direct consequence of quantum scarring. This protocol is in very close analogy with what has been done in[43,44], in which the many-body bitstring probabilities – essentially the same as our projection $Q$ – have been measured to detect genuine quantum effects. Indeed, accessing complex quantities not described by ETH, such as the bitstring probabilities, allows to search for new physics in spite of the thermalization of local observables, as we have proven for scarring. Our work opens many avenues for theoretical research, posing questions regarding the role of lattice geometry and interaction range, the fate of scarring in the limits $s \to \infty$ and $N \to \infty$, the effect of Hamiltonian terms beyond those in Eq. (1), and possibilities with other fermionic and bosonic

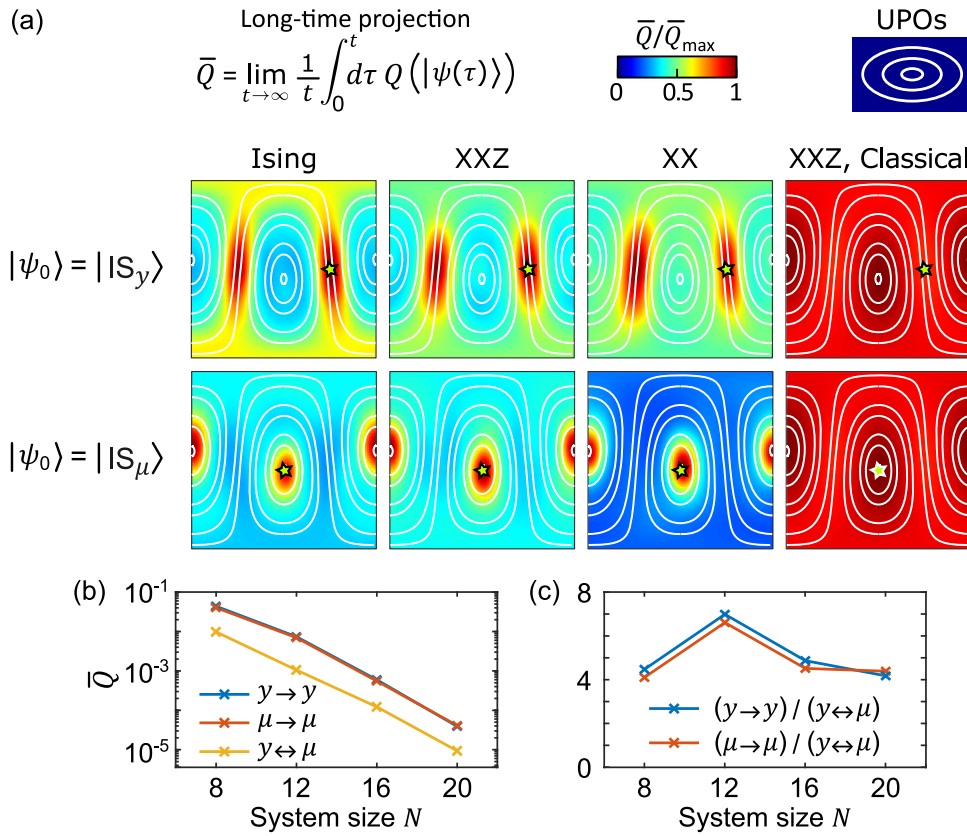

**Fig. 4 | Weak ergodicity breaking from quantum scarring. a** Time-averaged projection $Q(|\psi(t)\rangle)$ over the manifold of IS states, for various models and initial conditions. In the top row, the system is initialized in $|IS_y\rangle$, namely, on the IS UPO aligned along $\mathbf{y}$ ($\phi = \theta = \pi/2$, marked by a star), and is more likely to be found on the same UPO at long times. In the bottom row, the system is initialized in $|IS_\mu\rangle$, namely, on the IS UPO aligned along $\boldsymbol{\mu}$ (also marked by a star), and is more likely to be found there at long times. That is, the system retains some information on its initial condition and weakly breaks ergodicity. This is a quantum effect: the classical projection (see Supplemental Information), shown for the XXZ model, is insensitive to the initial condition and almost uniform. **b, c** Scaling of $\bar{Q}$ with system size $N$, with focus on the time-averaged return probabilities ($\bar{Q}(\mathbf{y})$ for $|\psi_0\rangle = |IS_y\rangle$, denoted $y \to y$, and $\bar{Q}(\boldsymbol{\mu})$ for $|\psi_0\rangle = |IS_\mu\rangle$, denoted $\mu \to \mu$) and cross probability ($\bar{Q}(\mathbf{y})$ for $|\psi_0\rangle = |IS_\mu\rangle$, and viceversa, denoted $y \leftrightarrow \mu$). While $\bar{Q}$ decays exponentially in $N$, scarring makes the return probabilities consistently larger than the cross probabilities, indeed by a factor $>4$ for all considered system sizes. In (**a**) we considered $s = \frac{1}{2}$, $N = 20$, and $\boldsymbol{\mu} = (2.4, 0, 0.4)$, and chose $\mathbf{J}$ ensuring that the system is fully thermal: $J_{zz} = -1.8$ for the Ising model, $J_{xx} = J_{yy} = -0.4$ and $J_{zz} = -1.8$ for the XXZ model, and $J_{xx} = J_{yy} = -1.4$ for the XX model. In (**b, c**) we considered the Ising model.

particles[15]. Realizing that the atypical eigenstates in so-called "many-body quantum scars" are in fact not scars is not merely a matter of terminology[31]: it opens the door to a whole new field of research, that of scars – genuine scars[11] – in many-body systems, and paves the way to a better understanding of the quantum-classical correspondence. We conclude by noting that, shortly after the completion of this work, the problem of genuine scarring in spin chains was also addressed in[58].

## Data availability
The authors declare that the data supporting the findings of this study are available within the paper and its supplementary information files.

## Code availability
The code used for the current study is available from the corresponding author on reasonable request.

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

## Acknowledgements

We thank A. Buchleitner, C. Castelnovo, A. Das, I. Ermakov, B. Fine, I. Kaminer, O. Lychkovskiy, S. I. Mistakidis, S. Moudgalya, A. Nunnenkamp, N. Rivera, and N. Yao for insightful discussions on related work and comments on the manuscript. A. P. acknowledges support by Trinity College Cambridge. C.B.D. was supported with the ITAMP grant No. 2116679.

## Author contributions

A.P. designed and performed the research. L.H.K. performed the study of $s > 1/2$. A.P., L.H.K., B.E., C.B.D. and J.K. took part in continued discussions and wrote the manuscript.

## Funding

## Competing interests

The authors declare no competing interests.
