## [Transparent Peer Review file · Nature Communications]

Genuine quantum scars in many-body spin systems

Corresponding Author: Dr Andrea Pizzi

Version 0:

Reviewer comments:

Reviewer #1

(Remarks to the Author)

The authors argue for an exponentially large family of quantum many-body scarred eigenstates in a generic set of spin chains that are well studied in condensed matter physics. In order to do this, they first identify a family of unstable periodic orbits in the large- S semiclassical limit of these spin chains and find a regime in parameter space where these unstable periodic orbits satisfy Heller's criterion, i.e. their Lyapunov time is much larger than their period, and are therefore favourable for quantum scarring. They then argue that the specific orbits they find "scar" the eigenstates of each spin- S Hamiltonian with approximately the same energy as the underlying classical orbit (viewed as a product state). Since there are generically exponentially many such eigenstates, the authors deduce that exponentially many eigenstates are scarred.

We find the first, classical part of this analysis original and interesting, particularly the identification of "interaction suppressing" orbits. However, while the idea that these orbits should cause scarring is plausible, we found the second part of this analysis much less convincing. In particular, we do not believe that the confident tone of the paper is justified given the weakness of the evidence presented for scarred eigenstates. For this reason, we think that before the paper is published, the claims made by the authors about the ubiquity of quantum scarring needs to be substantially strengthened.

Before getting into technical points, we think some comment is in order regarding the title and the "remark on nomenclature" on p. 1. First, the "remark": we agree with the authors that some of the early papers on quantum many-body scarring simply asserted a connection to Heller's work without any meaningful justification. However, it also seemed reasonably clear from the beginning that true Heller-type examples should exist and a minority of researchers have pursued these, see Ermakov et al. 2409.00258 or Hummel et al., PRL 130, 250402. We think citing these works and pointing out that the authors are continuing this line of inquiry would be more constructive than what is currently written. Second, the title of the paper is simply too general: for the reasons above, we do not think the viewpoint espoused by the authors is original enough to justify a title so similar in spirit to Ref. 26. Since they are not initiating this field, we think a more specific title would be appropriate, for example one that refers to spin chains (given that these are the only systems explicitly studied in the paper).

Regarding the analysis of quantum scarring, we had several specific questions and comments:

1. How is Q_{\max} defined? We would have thought that for a stationary classical configuration yielding an exact eigenstate, Q_{\max} would be equal to one. We could not find an explicit definition in the text and the values quoted on p.3 seem worryingly small to us from the viewpoint of experimental validation, or even statistical significance of these findings.
2. As far as we can tell, the only justification for the claim in the introduction of "exponentially many" scarred eigenstates is the evaluation of χ in the Supplement. However, χ is not a statistically meaningful quantity. A statistically meaningful test would require plotting the histogram of values of S beside the histogram for the random-state reference distribution considered by the authors. One would then have to show that the mean of S computed from ED for each N differs in a statistically significant way from the mean of the reference distribution, i.e. that the difference in means exceeds the standard error of the ED averaging for each N . Simply testing whether S exceeds some reference value, as in the current draft, is meaningless, because the excess could lie entirely within error bars.
3. We find most of the discussions non-universal in the sense that they pertain only to finite sizes. Perhaps all the effects the authors claim wash out in the thermodynamic limit (TDL)? For example in fig 3, how would the color plots change when

system size increases? Maybe the claimed relative differences wash out? It could be instructive to take two “distinct” points in the phase space and plot the value of Q vs system size i.e. finite size scaling --- do they go to 0 at different or distinct rates?

4. The color plot in Fig 3 of the classical projection seems rather arbitrarily drawn. Scrutinising the appendix, we see they consider a Gaussian wave packet around a point of the classical UPO, but the variance of this wave packet they just say is $\Delta \ll 1$, which seems arbitrarily chosen. (Note that in Truncated Wigner Approximation for the Wigner function to reproduce in leading order quantum dynamics of a spin- S system Δ should be taken to be $\sim 1/S$) Certainly there is some order of limits issue with Δ , N , observation time t ?

5. The authors claim fig S2 showing the phase space distribution of random states has no noteworthy structure. But we feel like there is! By eye it shows the relative fluctuations of random states on the phase space manifold are the same as the “quantum scars” they draw (We are not surprised of the feature that they do not lie on any particular UPO). So this makes the claimed case of ‘large’ relative scarring of the scars on UPOs less compelling. (We understand that the authors want to claim that the ‘largeness’ of the fluctuations should be compared relative to that produced from classical dynamics, but it is also valid to compare it to random quantum state).

6. How exactly is the “scarring” conjectured by the authors meant to be observed in a quantum simulator, as suggested in the abstract? The projection Q corresponds to a single wavefunction component, and a very small one at that, so we would like to know the grounds on which the authors interpret this as a “small albeit measurable correction” to ETH.

7. In general, we think that diagnosis of these scars via observables, rather than wavefunction components, would be more physically meaningful and more likely to be seen experimentally. However, it may be that fewer than exponentially many (if any) of the conjectured scarred eigenstates can be diagnosed via conventional observables.

To summarize, the interaction-suppressing classical orbits identified by the authors are new and interesting, and a robust quantum signature of these orbits would be a satisfying result. However, we think the authors fall short of establishing the latter and their analysis needs to be substantially strengthened and in fact made more quantitative, before the paper can be considered for publication in Nature Communications.

Reviewer #2

(Remarks to the Author)

Reviewer #3

(Remarks to the Author)

Please see the attached PDF file.

Reviewer #4

(Remarks to the Author)

Version 1:

Reviewer comments:

Reviewer #1

(Remarks to the Author)

This is a second report on “Genuine quantum scars in many-body spin systems” by Pizzi et al. First off, I want to express appreciation to the authors for taking the time and effort to reply very thoroughly to both sets of Referee reports.

I have thoroughly scrutinized the authors’ reply to my first report. I am glad that they have taken my suggestion and performed a quantitative analysis of the degree of scarring claimed as opposed to relying on an arguably loose and not too meaningful criteria as was done in the first version of the manuscript. The main new changes are encapsulated in Fig 3 and Fig 4.

Indeed, I agree with the authors that there is a clear quantitative difference (Fig 3a) between the distribution of (rescaled) overlaps between mid-spectrum eigenstates and Haar random states (and phase space states) versus IS states. I find the deviation from the Porter-Thomas deviation interesting.

Fig 3b shows also a difference of the distribution of the scarness parameter for mid-spectrum states compared to Haar

random states, showing that the mid-spectrum states do not behave like Haar random states. I too find this interesting.

The net take-home message is that there is enhancement of overlaps of quantum many-body eigenstates/dynamics on the UPOs of the classical model.

At the same time, as the authors themselves have pointed out, this effect is very weak, in the sense it goes down exponentially (as expected) with system size, so the enhancement is only relative (indeed supported by Fig 4).

The fact that the effect is weak is important experimentally: while I agree with the authors that bit-string probabilities can be measured in modern day quantum simulators nowadays, the exponentially small enhancement they are touting therefore needs an exponentially large number of measurements, limiting the observation of their effect to very small system sizes.

Thus, while I appreciate the message of the authors that there is a weak effect on the structure of many eigenstates, which is opposed to the standard way that quantum many-body scars have been discussed in the literature — that of a very strong effect (athermal entanglement, say) but for very few eigenstates, I am on the fence of whether this constitutes a novel enough result for a journal like Nat Comms. After all, there is no contradiction with the statements of ETH on quantum thermalization of local observables and the rate at which this equilibrium is achieved with system size. Note again I am not saying there are no new results in the paper: it is certainly curious to me the authors' observation of the enhancement on the UPOs — and I think this result should be published in some form — I am just unsure of the profoundness of this in the grand scheme of things. Perhaps the editors can seek another opinion.

Additional comments:

1. Before any form of publication, I would urge the authors to state clearly in the abstract that the enlarged weight they are pointing out is a very weak effect and is not contradictory to statements of ETH.

2. Also, in the inset of Fig 3b (and also the abstract), the authors state that the number of scarred eigenstates is 'large' and 'appears to grow exponentially with system size'. However, again like in my first report — I caution the authors against such hyperbolic statements: 'large' should always be in relation to the growth of some reference function. In this case, one should overlay the function $c2N$ on the inset, since $2N$ is the dimension of the full system size. I have plotted this on my computer. What I see is that the number of "scarred eigenstates", according to their criteria of scariness, seems a fixed ratio of the total number of eigenstates. From this point of view, this is then perhaps not too surprising: it just says mid-spectrum eigenstates do not behave like Haar random vectors, which we should expect anyway. After all, the comparison to Haar random vectors should only be reserved for coarse grained features like expectation values of local observables and their fluctuations, one cannot expect equality to Haar random vectors at all levels of features e.g. bit-string overlaps (I find the insistence of mid-spectrum eigenstates behaving like Haar random a loose, careless argument at best, and a strawman argument at worst).

Reviewer #3

(Remarks to the Author)

Please see the attached file.

Reviewer #4

(Remarks to the Author)

Version 2:

Reviewer comments:

Reviewer #1

(Remarks to the Author)

Reviewer #3

(Remarks to the Author)

Please see the attached file.

Reviewer #4

(Remarks to the Author)

Reply to the Referees

Note. In the revised manuscript, changes are highlighted using color and strikethroughs. In the following reply, when we quote the manuscript we mark the changes in light blue font.

Reviewer 1

We thank the Referee for their in-depth review and insightful comments, which helped us improve our manuscript. We address the Referee's report on a point-by-point basis below.

The authors argue for an exponentially large family of quantum many-body scarred eigenstates in a generic set of spin chains that are well studied in condensed matter physics. In order to do this, they first identify a family of unstable periodic orbits in the large- S semiclassical limit of these spin chains and find a regime in parameter space where these unstable periodic orbits satisfy Heller's criterion, i.e. their Lyapunov time is much larger than their period, and are therefore favourable for quantum scarring. They then argue that the specific orbits they find "scar" the eigenstates of each spin- S Hamiltonian with approximately the same energy as the underlying classical orbit (viewed as a product state). Since there are generically exponentially many such eigenstates, the authors deduce that exponentially many eigenstates are scarred.

We thank the Referee for their summary.

We find the first, classical part of this analysis original and interesting, particularly the identification of "interaction suppressing" orbits. However, while the idea that these orbits should cause scarring is plausible, we found the second part of this analysis much less convincing. In particular, we do not believe that the confident tone of the paper is justified given the weakness of the evidence presented for scarred eigenstates. For this reason, we think that before the paper is published, the claims made by the authors about the ubiquity of quantum scarring needs to be substantially strengthened.

We are glad that the Referee found the classical part of the analysis original and interesting. We agree with them that a more convincing case on the ubiquity of quantum scarring was needed. Having closely followed the Referee's feedback, in the revised manuscript and Supplemental Material we provide more evidence in support of our claims. The resulting paper is significantly strengthened, and we hope the Referee will be able to recommend it for publication in *Nature Communications*.

Before getting into technical points, we think some comment is in order regarding the title and the "remark on nomenclature" on p. 1. First, the "remark": we agree with the authors that some of the early papers on quantum many-body scarring simply asserted a connection to Heller's work without any meaningful justification. However, it also seemed reasonably clear from the beginning that true Heller-type examples should exist and a minority of researchers have pursued these, see Ermakov et al. 2409.00258 or Hummel et al., PRL 130, 250402. We think citing these works and pointing out that the authors are continuing this line of inquiry would be more constructive than what is currently written.

The Referee is absolutely right that the problem of genuine quantum scarring in many-body system was addressed in the *PRL* by Hummel *et al.*, which we inadvertently overlooked on this occasion. We very much align with their perspective on scars and we now cite them prominently. Their work indeed provides further motivation for ours. Quoting them, "In the context of the widely considered spin-chain-like systems such a study is, however, hampered by the fact that those quantum Hamiltonians do not have an obvious classical counterpart", a gap now filled by our work. The standing of our work within the literature is also clarified by the revised title (see next point), and by a new paragraph of the introduction in which we mention a few recent works on scarring in systems with all-to-all interactions, that are somewhere between single- and many-body.

As for the *arXiv* 2409.00258 by Ermakov *et al.*, note that our work (2408.10301) was in fact published *before* theirs. Hence, we have added the following concluding statement: "Note added: Shortly after the completion

of this work, the problem of genuine scarring in spin chains was also addressed in [58].”. Moreover, we have added the authors of that work, I. Ermakov, B. Fine, and O. Lyckkovskiy, to the acknowledgements (we had a few fruitful discussions with them after the publication of both papers on the *arXiv*).

Second, the title of the paper is simply too general: for the reasons above, we do not think the viewpoint espoused by the authors is original enough to justify a title so similar in spirit to Ref. 26. Since they are not initiating this field, we think a more specific title would be appropriate, for example one that refers to spin chains (given that these are the only systems explicitly studied in the paper).

We are happy to follow suggestion from the Referee and change the title in a more concrete: “**Genuine quantum scars in many-body spin systems**”.

Regarding the analysis of quantum scarring, we had several specific questions and comments:

1. How is Q_{\max} defined?

In short, Q_{\max} is the maximum value of Q within each plot, as we now clarify in the caption of Fig. 2: “**For each plot we define $Q_{\max} = \max_{\theta, \phi} Q(\theta, \phi)$.**”.

As a more general remark, note that plotting Q or Q/Q_{\max} is equivalent, but the former requires to show a colorbar with complicated values (decimal points, powers of 10, etc) for each plot. This would result in very busy and messy figures (e.g., in Fig. 2 one would need 14 such colorbars). For the sake of a cleaner figure, and because the figure aims anyway at emphasizing the relative contrast of Q within the (θ, ϕ) plane, we prefer to plot Q/Q_{\max} and effectively get rid of dozens of colorbars and tags. Note that the same stylistic choice is implicitly done when using a contour plot, like in Heller’s original paper [*PRL* **53**.16, 1515 (1984)]: also in this case the representation of the eigenstates focuses on conveying the qualitative, “visual” features of Q , without specifying Q_{\max} (see example in Fig. R1).

FIG. R1. **Pcolor vs contour.** For the traditional example of a billiard, the wavefunction of the eigenstates $|\psi_n\rangle$ (namely $Q = |\langle \mathbf{r} | \psi_n \rangle|^2$) can be plotted using either a color plot (top, e.g., `pcolor` in MATLAB), or a contour plot (bottom, as in Heller’s 1984 *PRL*). Plotting `pcolor(Q/Qmax)` is morally equivalent to plotting `contour(Q)`: in both cases the focus is on the qualitative features of Q and the n -dependent Q_{\max} is not specified.

1 - [cont’d]. *We would have thought that for a stationary classical configuration yielding an exact eigenstate, Q_{\max} would be equal to one.*

If a stationary classical configuration yielded an exact eigenstate, then indeed one would expect $Q_{\max} = 1$. However, there is no reason to expect that the stationary classical configuration should yield an exact eigenstate. In our case, as in Heller’s billiard, fixed points and periodic orbits cease to be such once quantum fluctuations are added, no matter how small they are, because the chaotic dynamics tends to spread these fluctuations across phase space. The beauty of scarring is that some eigenstates can, *against this intuition*, retain an enhanced weight on the classical unstable fixed points and periodic orbits, but it is not expected that some exact and fully localized eigenstates should exist (the situation changes substantially if we sit in a regular region of the phase space, but this is not relevant for our work nor for Heller’s).

1 - [cont’d]. *We could not find an explicit definition in the text and the values quoted on p.3 seem worryingly small to us from the viewpoint of experimental validation, or even statistical significance of these findings.*

We have now added an explicit definition to the caption of Fig. 2, see the quote in the reply above. We agree

that Q_{\max} takes small values. This is in a way comforting – a large value of Q_{\max} would have contradicted established paradigms such as the ETH not just for one specific model, but for a whole family of popular models [Eq. (1)], which is implausible. Our claim is that, while small in absolute terms, Q_{\max} is large in relative terms (we will elaborate on this in reply to point 6 below). While not contradicting ETH, this is enough to profoundly challenge our understanding of many-body quantum chaos and to be measurable in experiments.

2. As far as we can tell, the only justification for the claim in the introduction of “exponentially many” scarred eigenstates is the evaluation of χ in the Supplement. However, χ is not a statistically meaningful quantity. A statistically meaningful test would require plotting the histogram of values of S beside the histogram for the random-state reference distribution considered by the authors. One would then have to show that the mean of S computed from ED for each N differs in a statistically significant way from the mean of the reference distribution, i.e. that the difference in means exceeds the standard error of the ED averaging for each N . Simply testing whether S exceeds some reference value, as in the current draft, is meaningless, because the excess could lie entirely within error bars.

We thank the Referee for their valuable suggestions, that we have followed with a successful outcome. We now consider the full distribution of S , show that it differs significantly for eigenstates and Haar random states, and slightly adapt the definition of the scarring probability χ and of the number of scarred eigenstates N_{scarred} to make them more statistically meaningful and consistent. This analysis led to some major revisions.

In the main text, we have added a new paragraph and figure:

“[...] we introduce a “scarness” parameter $S = 4D \times \max_{\text{IS}} \oint Q_{\psi}$, where $\oint Q$ denotes averaging of Q along each UPO, and \max_{IS} maximization over the IS UPOs (details in [43]). Sampling $|\psi\rangle$ yields a probability distribution for S , shown in Fig. 3(b). While the mid-spectrum eigenstates are often compared to Haar random states, they yield a distribution of the scarness S with a much fatter tail. Indeed, the S of many eigenstates is larger than the S of most Haar states, and the number of scarred eigenstates N_{scarred} grows exponentially with the considered system sizes N (inset).”

FIG. R2. (extract from Fig. 3) “**Quantitative features of scarring.** [...] (b) The distribution of the scarness parameter S for mid-spectrum eigenstates (blue) has a long tail compared to Haar random states (red). Indeed, the number N_{scarred} of scarred eigenstates, defined as the number of eigenstates with S larger than the 90-th percentile of the Haar states (red dashed line), minus 10% [43], is large and appears to grow exponentially with system size (inset). In (a) and (b) the eigenstates are sampled from the middle of the spectrum, namely among the 10% eigenstates with lowest $|E\rangle$. [...] Here, $N = 20$ except where otherwise and explicitly specified. All other parameters are as in Fig. 2.”

More details are presented in the revised Supplementary Information:

“As mentioned above, scarring manifests in Q having large fluctuations and having a structure in phase space that reflects the underlying UPOs. To quantify both effects at once, we consider the average Q over an UPO, which we denote $\oint_{\text{UPO}} Q$, and look for the UPO that maximizes it. For a wavefunction $|\psi\rangle$ with projection Q we thus introduce the following figure of merit for scarring:

$$S = 4D_{\nu} \times \max_{\text{UPOs}} \oint_{\text{UPO}} Q, \quad (\text{S5})$$

where \mathcal{D}_ν is the size of the considered special symmetry sector and the factor 4 compensates for the fact that the IS states have weight 1/4 on it (these normalization factors are anyway arbitrary and play no role in the following analysis). We compute S for the mid-spectrum eigenstates and for an ensemble of Haar random states, which serves as a benchmark. The probability distribution of the resulting S is shown in the main Fig. 3(b). While the mid-spectrum eigenstates of quantum chaotic Hamiltonians are often compared to Haar random states, and while the distributions obtained for the two indeed have some overlap, the distribution for the eigenstates is shifted towards larger values of S compared to that of the Haar random states. For instance, for $s = 1/2$, $N = 20$, and sector 0^+ we find that $\langle S \rangle_E - \langle S \rangle_H \approx 1.98 \sqrt{\langle S^2 \rangle_H - \langle S \rangle_H^2}$, where $\langle \dots \rangle_E$ and $\langle \dots \rangle_H$ denote average over the ensembles of mid-spectrum eigenstates and Haar random states, respectively. That is, the average S for the eigenstates is two standard deviations larger than the average S for the Haar random states.

This mismatch can be used to quantify the number of scarred eigenstates. To this end we consider the N_e mid-spectrum eigenstates, say N'_e the number of them that have S larger than the 90-th percentile of the S for the Haar random states, and introduce the scarring probability χ as

$$\chi = \frac{N'_e}{N_e} - 0.1, \quad (\text{S6})$$

where the -0.1 compensates for the fact that even for random Haar states S is larger than the 90-th percentile 10% of the times (by definition of 90-th percentile). If the eigenstates behaved like Haar random states, one would get $\chi = 0$. But because, instead, many of the eigenstates have S larger than most of the Haar random states, the scarring probability χ can then take relatively large values ($\sim 40\%$ for the considered parameters), indicating that a significant fraction of the eigenstates is scarred by the IS UPOs. Analogously, we can define the number of scarred mid-spectrum eigenstates as

$$N_{\text{scarred}} = N'_e - 10\%N_e = \chi N_e, \quad (\text{S7})$$

which we plotted in the inset of Fig. 3(b) in the main text.

Note that our analysis has focussed on the mid-spectrum eigenstates, the number of which, N_e , depends on how wide the central energy window is chosen, which is arbitrary. The N_{scarred} that we considered does not count the eigenstates outside of such energy window. Because the non-mid-spectrum eigenstates could also be scarred, one might say that N_{scarred} underestimates the number scarred eigenstates. At the end of the day, quantifying scarring is not an easy task, and there is no unique way of doing it. The analysis above succeeds in indicating that scarring affects many eigenstates, but for instance does not consider that certain eigenstates are scarred by multiple UPOs. There is thus room for a more systematic quantitative analysis of scarring in many-body systems, which goes however beyond the scope of this work.”

In particular, going back to the Referee’s remark, focusing on the 90% percentile ensures the statistical significance of our results, because it means that we only count the eigenstates with S larger than *most* of the Haar random states. Even with such a high threshold, we find that χ is large. For instance, for $N = 20$ and $s = 1/2$ we find that the average of S over the eigenstates exceeds the average of S over Haar random states by ≈ 2 standard deviations.

3. We find most of the discussions non-universal in the sense that they pertain only to finite sizes. Perhaps all the effects the authors claim wash out in the thermodynamic limit (TDL)? For example in fig 3, how would the color plots change when system size increases? Maybe the claimed relative differences wash out? It could be instructive to take two “distinct” points in the phase space and plot the value of Q vs system size i.e. finite size scaling — do they go to 0 at different or distinct rates?

We thank the Referee for these remarks. We have followed thir suggestions and added a scaling analysis to Fig. 4(b,c), which we report here:

“A scaling analysis is presented in Fig. 4(b,c), showing that \bar{Q} decays exponentially with N while its relative enhancement does not, exhibiting a memory effect.”

FIG. R3. (extract from Fig. 4) “[...] (b,c) Scaling of \bar{Q} with system size N , with focus on the time-averaged return probabilities ($Q(\mathbf{y})$ for $|\psi_0\rangle = |\text{IS}, \mathbf{y}\rangle$ and $Q(\boldsymbol{\mu})$ for $|\psi_0\rangle = |\text{IS}, \boldsymbol{\mu}\rangle$) and cross probability ($Q(\mathbf{y})$ for $|\psi_0\rangle = |\text{IS}, \boldsymbol{\mu}\rangle$, and viceversa). While \bar{Q} decays exponentially in N , scarring makes the return probabilities consistently larger than the cross probabilities, indeed by a factor > 4 for all considered system sizes.”

In addition to this clarification, new scaling analyses have also been added to the inset of the new Fig. 3 (see reply to point 2 above and 5 below).

While it is not easy to make ultimate statements on the $N \rightarrow \infty$ limit (a common situation, as for MBL or even scarring in the PXP model), numerics suggests that washing out of our findings in such limit is at the very least not at all obvious. Most importantly, however, we should emphasize that the relevance and interest of our results is anyway independent of the $N \rightarrow \infty$ limit.

To explain, let us consider the original example of scarring in billiards. The ground and first excited states, shown in Fig. R4, appear rather regular, similar to those of a rectangular box potential – chaos is not really allowed to manifest at such low energies. More interesting are the highly excited states of the semiclassical regime: for them chaos is a priori allowed (indeed expected), and this is why scarring, namely a form of suppression of chaos, is so surprising and special. Semiclassicality is a handle to enable chaos, bringing us into a regime where seeing structure in the form of scars is exciting. While it is interesting to study what happens in the $\hbar \rightarrow 0$ limit, this is not necessary: Heller’s 1984 work, at a *small but finite* \hbar , is already interesting, because it defeats the expectation of chaos (e.g., Berry’s conjecture). Sure, Heller could have gone to smaller \hbar , and found (as done later [e.g., *PRL* **61**.20, 2288 (1988)]) that scarring becomes rarer, but that is not the key point. Indeed, we would argue that scarring in billiards is studied in the semiclassical *regime* ($\hbar \ll 1$), not in the semiclassical *limit* ($\hbar \rightarrow 0$), which is not necessary and anyway computationally intractable.

FIG. R4. **Semiclassicalness as a handle to chaos in a billiard.** The wavefunction $|\psi_n(x, y)|^2$ of a few eigenstates of the Bunimovich billiard is shown. The ground and first five excited states ($n = 1, 2, \dots, 6$) appear rather regular. Chaos can manifest only at larger energies, i.e., in the semiclassical limit (e.g., for $n = 5056$) and be to some extent suppressed by quantum scarring (e.g., for $n = 5058$). Here, we consider a billiard that is twice as wide than high.

The same logic extends to the many-body setting upon replacing \hbar with $1/N$. A large enough N is required for a notion of quantum chaos to be possible and indeed expected, setting a standard (e.g., level statistics and ETH) against which deviations (e.g., the memory effect in Fig. 4) are surprising. Instead of working in the thermodynamics *limit* $N \rightarrow \infty$, we just need to work in the many-body *regime* $N \gg 1$, so that a notion of chaos is possible and its failure interesting. Thus, while our scaling analyses suggests that some of our results could extend to $N \rightarrow \infty$, this limit is not necessary for a meaningful discussion of many-body scarring.

4. The color plot in Fig 3 of the classical projection seems rather arbitrarily drawn. Scrutinising the appendix, we see they consider a Gaussian wave packet around a point of the classical UPO, but the variance of this wave packet they just say is $\Delta \ll 1$, which seems arbitrarily chosen. (Note that in Truncated Wigner Approximation for the Wigner function to reproduce in leading order quantum dynamics of a spin- S system Δ should be taken to be $\sim 1/S$) Certainly there is some order of limits issue with Δ , N , observation time t ?

We thank the Referee for this remark. The fluctuations Δ of the ensemble are indeed pretty arbitrary, and for a good reason. Our main claim is that the time-averaged projection \bar{Q} (which is obtained for $t \rightarrow \infty$) does not depend on the initial condition for classical systems, whereas it does for quantum ones. This claim can be made for various Δ and N . The smaller Δ and the longer it takes for the classical wavepacket to scramble across phase space and “forget” its initial condition, but this will always occur eventually, over a Ehrenfest time $\sim \frac{1}{\lambda} \log \Delta$ (unless the limit $\Delta \rightarrow 0$ is taken before the limit $t \rightarrow \infty$, but there is no reason to take a limit $\Delta \rightarrow 0$ in the first place). In the Truncated Wigner Approximation, Δ is carefully chosen to match the quantum fluctuations at short times, but this is not our goal here, and the choice of Δ is thus pretty arbitrary. Indeed, in Fig. R5 below we show that \bar{Q}_c does not depend on Δ .

FIG. R5. **Independence of \bar{Q}_c from Δ .** We reproduce the results in Fig. 3 (Fig. 4 in the revised manuscript), but for various values Δ of the fluctuations in the ensemble of initial conditions. The results, and in particular the claim that \bar{Q}_c is the same for the two considered initial conditions, do not depend on Δ .

In the revised Supplementary Information, we have clarified these points in the following passage:

“A perturbation of this point is obtained by rotating each spin by an angle θ_j drawn at random from a Gaussian distribution with standard deviation $\Delta = 10^{-5}$ (almost identical results are obtained for any $\Delta \ll 1$, provided that the time averaging is performed over a time $\gg \frac{1}{\lambda} \log \Delta$).”.

5. The authors claim fig S2 showing the phase space distribution of random states has no noteworthy structure. But we feel like there is! By eye it shows the relative fluctuations of random states on the phase space manifold are the same as the “quantum scars” they draw (We are not surprised of the feature that they do not lie on any particular UPO). So this makes the claimed case of ‘large’ relative scarring of the scars on UPOs less compelling. (We understand that the authors wan to claim that the ‘largeness’ of the fluctuations should be compared relative to that produced from classical dynamics, but it is also valid to compare it to random quantum state).

We thank the Referee for this very insightful comment, that pushed us to a much deeper analysis and understanding.

We agree with the Referee that Fig. S2 does not show that the fluctuations of Q for a random state are smaller than for an eigenstate – this was indeed not the goal of such figure. Instead, that figure emphasizes that the projection of a random state does not reflect the underlying classical trajectories, which is what we meant by “it has no noteworthy structure”. While maybe not too surprising (the random states are agnostic of the Hamiltonian, so how could they possibly reflect its classical trajectories?), we thought it would be worth showing, so we could emphasize the striking contrast with the eigenstates. We have clarified by slightly rewording the passage: “Random states do not show any noteworthy structure **in relation to the underlying classical UPOs.**”.

This said, the Referee is rightly interested in whether, going beyond the visual and qualitative features of Fig. S2, the projection Q shows quantitative differences between eigenstates and random states. The answer is affirmative: the projection has much larger fluctuations for the eigenstates than for the random states. We have now made this statement precise with a new paragraph and figure in the main text:

“Having seen in Fig. 2 the visual, qualitative features of scarring, we now turn to a quantitative analysis, showing that the eigenstates are anomalously large on the UPOs. We consider the mid-spectrum eigenstates $|E_n\rangle$ of the symmetry sector with zero momentum and left-right mirror parity $+1$, and overlap them with

three types of states $|\psi\rangle$: Haar random states, phase-space states $\{|\mathbf{s}_i\rangle\}$, and IS states. To make sure these are treated on par, namely that generic phase-space states are not penalized for being less symmetric than the IS states, we renormalize all the states to have weight 1 on the considered symmetry sector (details in [43]). That is, we consider the rescaled projection of $|E_n\rangle$ on $|\psi\rangle$, namely $x = \mathcal{D} \frac{|\langle E_n | \psi \rangle|^2}{\langle \psi | \hat{U} \hat{U}^\dagger | \psi \rangle}$, where \mathcal{D} is the size of the considered symmetry sector and \hat{U}^\dagger the operator projecting on it. Sampling $|E_n\rangle$ and $|\psi\rangle$ yields a probability distribution for x , shown in Fig. 3(a). Haar random states yield the Porter-Thomas distribution e^{-x} [48,49], setting a benchmark for quantum chaotic behaviour. This benchmark is closely followed by the projections of the eigenstates on generic points of the phase space $\{|\mathbf{s}_i\rangle\}$. A deviation from the benchmark is found for projections on the IS states, yielding a fat tail in the distribution of x . This is remarkable: it shows that, due to scarring, the eigenstates can indeed have an anomalously large projection on the UPOs.”

FIG. R6. (extract from Fig. 3.) “**Quantitative features of scarring.** (a) Distribution of the rescaled overlaps x between mid-spectrum eigenstates and various families of states. The overlaps with Haar random states (red) have distribution e^{-x} (dashed), similarly to the overlaps with generic points $\{|\mathbf{s}_i\rangle\}$ of the phase space (green). By striking contrast, the overlaps with the IS states (blue) yield a much fatter tail, that is, scarring makes some eigenstates anomalously large on the UPOs. [...] In (a) and (b) the eigenstates are sampled from the middle of the spectrum, namely among the 10% eigenstates with lowest $|E|$. In (a), the phase-space states are obtained sampling the spins $\{s_i\}$ uniformly and independently from the sphere, and the IS states are obtained sampling \mathbf{s}_1 uniformly from the sphere and alternating the other spins at every other site as in Fig. 1(a). Here, $N = 20$ except where otherwise and explicitly specified. All other parameters are as in Fig. 2. ”

Further details, and in particular a necessary careful account of symmetries, are provided in the revised Supplementary Information:

“Quantitative features of scarring: details and role of the symmetries

We now turn to the question how to quantify scarring, and in particular whether the eigenstates have an anomalously large projection Q on the UPOs. This is nontrivial, because the meaning of “large” is subtle: just out of random fluctuations, even the projection of a Haar random wavefunction happens to be somewhat larger in some points of the phase space, as in Fig. S2, and indeed have some finite probability to be arbitrarily close to the maximum value 1. In what sense then, exactly, can the projection of a scarred eigenstate on a UPO be said “anomalously large”?

The first and mandatory step to sensibly address these questions is to analyze the symmetries of the problem. The system is invariant under translation, which splits the Hamiltonian in sectors with a well defined momentum $k = \frac{2\pi}{N}n$, with $n = 0, 1, 2, \dots, N-1$. Due to the left-right mirror reflection symmetry, the sectors with $k = 0$ and $k = \pi$ can be further divided into blocks with parity $P = \pm 1$. The symmetry sectors can thus be tagged by their momentum k and, if $k = 0$ or $k = \pi$, by a superscript \pm indicating the parity. Let us call \hat{U}_ν the operator that projects from the full Hilbert space to the ν -th sector. We denote \mathcal{D}_ν the size of the ν -th sector, and $\mathcal{D} = \sum_\nu \mathcal{D}_\nu$ the total size of the Hilbert space. For instance, for $s = 1/2$ and $N = 20$ we find $\mathcal{D}_{0^+} = 27012$, $\mathcal{D}_{\pi^+} = 25984$, $\mathcal{D}_{0^-} = 25476$, $\mathcal{D}_{\pi^-} = 26496$, $\mathcal{D}_{\pi/2} = 52380$, and $\mathcal{D} = 1048576$.

It is easy to verify that the TI states are fully contained within the 0^+ sector while the IS states distribute their weight equally within four “special” sectors 0^+ , π^+ , $+\frac{\pi}{2}$, and $-\frac{\pi}{2}$. It follows that one can talk about

scarring from the TI UPOs only for the eigenstates of the sector 0^+ , and scarring from the IS UPOs only for the eigenstates of the sectors 0^+ , π^+ , $+\frac{\pi}{2}$, and $-\frac{\pi}{2}$. [...]

These symmetry considerations done, let us go back to the question whether the projection of the eigenstates on the UPOs is “large”, focussing on one of the four special sectors. We consider the 10% most central (i.e., with smallest $|E_n|$) eigenstates of the sector. Let us call N_e the number of such mid-spectrum eigenstates, for instance $N_e = 2701$ for $s = 1/2$, $N = 20$, and sector $\nu = 0^+$. As in Fig. 3 in the main text, to quantify scarring we want to compare the overlap of the mid-spectrum eigenstates with various families of states. The first are Haar random states, that should be sampled within the symmetry sector of interest. The second are generic phase space states $|\{\mathbf{s}_i\}\rangle$, drawn upon sampling the spins $\{\mathbf{s}_i\}$ uniformly and independently from the surface of a unit sphere. The third and last are IS states, which are also of the form $|\{\mathbf{s}_i\}\rangle$, but for which \mathbf{s}_1 is sampled uniformly from the surface of a sphere while all the other spins are set following the IS condition, namely $\mathbf{s}_i = \nu_i \mathbf{s}_1$ with $\nu_i = (+ + - - + + - - \dots)$. Crucially, generic phase-space states $|\{\mathbf{s}_i\}\rangle$ lack any symmetry, and their weight is thus distributed among all symmetry sectors, e.g., only $\approx 1/(2N)$ -th of their weight is on the 0^+ sector on average. By contrast, the IS states are highly symmetrical and have a relatively large weight $1/4$ on each of the special symmetry sectors. Just out of symmetry arguments it follows that an eigenstate of one of the special sectors will tend to have a larger overlap with an IS state than with a generic phase space state, on average by a factor $\sim \frac{1/4}{1/(2N)} = \frac{N}{2}$ for sectors 0^+ and π^+ , and by a factor $\sim \frac{1/4}{1/(N)} = \frac{N}{4}$ for sectors $+\frac{\pi}{2}$ and $-\frac{\pi}{2}$. Of course, such an enhancement is first of all an effect of the symmetries, not of scarring.

To distill the enhancement due purely to scarring, we effectively cancel the effect of the symmetries by projecting out the part of the wavefunction that is not in the symmetry sector of interest. That is, given a state $|\psi\rangle$ (e.g., a phase-space state or a IS state), we define the symmetrized state

$$|\psi\rangle_\nu = \frac{\hat{U}_\nu^\dagger |\psi\rangle}{\sqrt{\langle \psi | \hat{U}_\nu \hat{U}_\nu^\dagger | \psi \rangle}}, \quad (\text{S3})$$

that, by construction, has weight 1 on the symmetry sector of interest ν , and 0 on the other sectors. Thus, while phase-space states and IS states have on average a different weight on ν , their symmetrized versions in Eq. (S3) by construction have the same weight 1, and can thus be compared fairly. Furthermore, now that all the states of interest have weight 1 on the considered sector ν , we can also conclude that their average weight on the basis states of such sector is \mathcal{D}_ν^{-1} . For a sector of interest $\nu \in (0^+, \pi^+, +\frac{\pi}{2}, -\frac{\pi}{2})$, an eigenstate $|E\rangle_n$ of such sector, and a state $|\psi\rangle$, we can thus define the rescaled overlap

$$x = \mathcal{D}_\nu |\langle E_n | \psi \rangle_\nu|^2 = \mathcal{D}_\nu \frac{|\langle E_n | \psi \rangle|^2}{\langle \psi | \hat{U}_\nu \hat{U}_\nu^\dagger | \psi \rangle}, \quad (\text{S4})$$

that is what we have used in the main text for the case of $\nu = 0^+$. Specifically, to quantify the overlap of the eigenstates with a certain family of states (Haar, phase-space states, or IS states), we generate an ensemble of rescaled projections x upon sampling both $|E_n\rangle$ (from the mid-spectrum eigenstates) and $|\psi\rangle$ (from the specific family of states). For each family of states this results in a probability distribution for x , which we have plotted in Fig. 3. ”

6. How exactly is the “scarring” conjectured by the authors meant to be observed in a quantum simulator, as suggested in the abstract? The projection Q corresponds to a single wavefunction component, and a very small one at that, so we would like to know the grounds on which the authors interpret this as a “small albeit measurable correction” to ETH.

We thank the Referee for this question, that motivated us to clarify and provide more context in the revised manuscript. First of all, directly measuring the many-body eigenstates is of course not realistically possible, and scarring would thus have to be indirectly measured via its effect on the dynamics, as in Fig. 4. A few considerations are due:

(i) The full “tomography” of Fig. 4, in which θ and ϕ are used to scan the manifold of UPOs, would not be necessary. To prove the point that the system remembers its initial condition, namely that \bar{Q} is larger

on the orbit on which the system was initialized, it would be enough to consider one initial condition and the projection on two points in phase space (one on the orbit, \bar{Q}_{on} , and one not, \bar{Q}_{off}), for a total of just 2 evaluations of \bar{Q} ;

(ii) The projection \bar{Q} would be measured from repeated experiments. Because \bar{Q} is exponentially small in N , the number of repetitions is exponentially large. However, as we have argued in reply to point 3 above, there is no need to push N to be as large as possible. For instance, a meaningful experiment to prove scarring could already be done for $N = 12$, for which $\bar{Q}_{\text{max}} \sim 10^{-2}$ and $10/\bar{Q}_{\text{max}} \sim 10^3$ repetitions should suffice, which is reasonable;

(iii) Measuring a local observable $\langle \hat{O} \rangle = 0.1$ with precision 10^{-3} would be hard due to noise thresholds. However, while the *absolute* value of \bar{Q} is small, its *relative* fluctuations are large [e.g., $\bar{Q}_{\text{on}}/\bar{Q}_{\text{off}} \sim 4$ in Fig. 4(c)]. Thus, even if noise and imperfections would introduce a large uncertainty in \bar{Q} (say, pessimistically, 50%), the contrast between \bar{Q}_{on} and \bar{Q}_{off} , hence scarring, would still be detected;

(iv) Indeed, and most importantly, similar measurements have already been done! Something that we only recently came to appreciate is that Google’s famous quantum supremacy experiment [Arute *et al.*, *Nature* **574**, 505–510 (2019)] deals with essentially the same situation: they initialize a many-body wavefunction, time evolve it, project it on bitstrings, average over many runs, compute the (exponentially small) bitstring probabilities, and argue for genuine quantum features in it. Indeed, our $Q(\theta, \phi)$ is nothing but a bitstring probability, just with respect to a rotated basis, which can be obtained by applying suitable and simple one-bit unitaries right before measuring. It should thus be possible to verify our findings in ever improving state of the art quantum simulators and computers, with a protocol in very close analogy with Google’s from six years ago.

In the following passage of the revised manuscript we now better contextualize and motivate our claim on the feasibility of an experiment:

“In particular, by repeatedly preparing a product state, letting it evolve, and measuring it in a different basis, one should be able to show that the system is more likely to be found on the UPO it was prepared on (Fig. 4), which is a direct consequence of quantum scarring. This protocol is in very close analogy with what has been done in [49,57], in which the many-body bitstring probabilities – essentially the same as our projection \$Q\$ – have been measured to detect genuine quantum effects.”

7. In general, we think that diagnosis of these scars via observables, rather than wavefunction components, would be more physically meaningful and more likely to be seen experimentally. However, it may be that fewer than exponentially many (if any) of the conjectured scarred eigenstates can be diagnosed via conventional observables.

Scarring appears indeed to leave a mark on the dynamics of local observables. However, compared to the effects on the wavefunction components, the effects on simple observables is less striking and likely harder to measure, which is ultimately due to thermalization.

Let us elaborate with an example. In Fig. R7 we show the dynamics of a simple local observable, $\langle \hat{\sigma}_1^y \rangle(t)$, starting from a state on a IS UPO (namely the IS state with $\langle \hat{\sigma}_1^y \rangle(0) = 1$). At short times the system tends to precess around the magnetic field $\boldsymbol{\mu}$, leading to oscillations of $\langle \hat{\sigma}_1^y \rangle(t)$ at frequency $\omega_{\text{upo}} = |\boldsymbol{\mu}|$, but is strongly damped by the spreading of correlations and the onset of thermalization. The short-time damped oscillations are not surprising nor directly indicative of scarring: the intuition behind them is ultimately classical, and indeed they are well captured by a classical simulation within the discrete Truncated Wigner Approximation (blue lines in Fig. R7). Instead, the “anomalous” feature due to scarring is that oscillations at frequency $\sim \omega_{\text{upo}}$ persist in the quantum case while they cease in the classical case (as it would happen in billiards). The amplitude of these fluctuations is exponentially small in N , in agreement with thermalization and ETH.

The anomalous oscillations in $\langle \hat{\sigma}_1^y \rangle(t)$ would be hard to measure in experiments, due to noise. Instead, the anomaly in the wavefunction components is striking: from the analysis of \bar{Q} in Fig. 4 we see that it is ~ 4 times more likely to find the system in $|IS, \mathbf{y}\rangle$ than in $|IS, \boldsymbol{\mu}\rangle$, for all system sizes considered. The probabilities to find the system in a given bitstring are small in absolute terms, which would require running exponentially many experiments, but the anomaly is large in relative terms, meaning that it should be

observed despite noise, an argument which echoes from point (iii) in our reply to (6) above.

In general, we find that the effects of scarring on local observables are less striking than those on the wavefunction components. Indeed, the former is limited by ETH whereas the latter is not. ETH sets a stringent limit on what can be expected for local observables in non-equilibrium many-body systems, which is why nontrivial phenomena are usually sought in ETH-violating systems. Modern quantum simulators are making it increasingly possible to access more complicated observables, such as wavefunction components in Refs. [49,57]. Not described by ETH, these quantities open the way to new physics even in generic systems, a message that our work establishes. That is, new and exciting physics can be present even in the absence of nonthermal eigenstates. In the revised outlook, we emphasize this point in the following passage:

“Indeed, accessing complex quantities not described by ETH, such as the bitstring probabilities, allows to search for new physics in spite of the thermalization of local observables, as we have proven for scarring.”

FIG. R7. **Dynamics of local observables.** We compare the classical and quantum dynamics of a simple local observable $\langle \hat{\sigma}_1^y \rangle$ starting from the IS state aligned along the \mathbf{y} axis. The top row shows the real time dynamics, whereas the bottom row its Fourier transform. System sizes $N = 8, 12, 16$ are considered in the left, middle, and right columns, respectively.

Technical details: The classical results are obtained averaging over 10^6 trajectories within the discrete Truncated Wigner Approximation (d-TWA) [PRX 5.1, 011022 (2015)]. The Fourier transform is obtained Fourier transforming $\langle \hat{\sigma}_1^y \rangle$ within a time window $[t_0 - 1.5T_{\text{UPO}}, t_0 + 1.5T_{\text{UPO}}]$, averaging over $t_0 \in [4.5T_{\text{UPO}}, 12.5T_{\text{UPO}}]$, and normalizing in such a way that a perfectly sinusoidal precession around the magnetic field $\boldsymbol{\mu}$ would yield an intensity 1 at frequency ω_{UPO} . Here, T_{UPO} is the period of the IS UPO and $\omega_{\text{UPO}} = \frac{2\pi}{T_{\text{UPO}}}$ its frequency. Besides N , all other parameters are as in Fig. 2.

To summarize, the interaction-suppressing classical orbits identified by the authors are new and interesting, and a robust quantum signature of these orbits would be a satisfying result. However, we think the authors fall short of establishing the latter and their analysis needs to be substantially strengthened and in fact made more quantitative, before the paper can be considered for publication in Nature Communications.

We genuinely thank the Referee(s) for their detailed, thoughtful, and constructive comments and criticism. Addressing them has resulted in an expanded, more quantitative, and significantly strengthened paper, which we hope the Referee(s) will now recommend for publication in *Nature Communications*.

Reviewer 2

The Reviewer has co-reviewed the manuscript with one of the reviewers who provided the listed reports.

Reviewer 3

In this paper, the authors discuss the existence of quantum scars in many-body systems. Unlike the so-called “quantum many-body scars,” which are atypical energy eigenstates that break eigenstate thermalization hypothesis, their quantum scars are rooted in unstable periodic orbits (UPOs) in a certain classical limit and regarded as a small correction to typical thermal eigenstates. To demonstrate this, they consider general spin- s many-body systems of N spins with nearest-neighbor interactions. For a classical limit, they consider $s \rightarrow \infty$ and obtain the corresponding classical nonlinear equation, which generally exhibits chaos and possesses unstable periodic orbits. Then, they demonstrate that such UPOs lead to quantum scars for fully quantum case ($s = 1/2$). In particular, by calculating the projection of the wave function onto the classical phase space, they find that the thermal energy eigenstates are scarred, i.e., their distributions have patterns reflecting the shape of the UPOs of the corresponding classical limits. Then, they discuss how the scars affect the long-time dynamics of isolated quantum systems, mentioning the distinctions from the classical counterpart. Finally, they conclude the paper with addressing the relevance to experiments.

We thank the Referee for their accurate summary.

In my opinion, the topic of the manuscript is interesting and worth consideration of Nature Communications. Indeed, in contrast with previous studies of quantum many-body scars, the scars considered here are in line with the original motivation of scars discussed in semiclassical systems such as quantum billiards [11]. While spin 1/2 systems do not have a well-defined classical limit, the paper shows that the classical models with $s \rightarrow \infty$ are relevant even for such systems, which I feel is surprising. While quantum scars found in this manuscript have less effects to thermalization dynamics, they provide an important clue to understand quantum many-body chaos and quantum-classical correspondence. Also, I think that the manuscript is clearly written and easy to read for non-specialists.

We are glad that the Referee found our the manuscript interesting, worth consideration in *Nature Communications*, to provide important clues to understand many-body quantum chaos and the quantum-classical correspondence, and well written.

That being said, I hesitate to recommend the manuscript for publication in the current form since I have several concerns. First, I feel that some statements do not seem sufficiently supported, which should be modified. Second, the novelty compared with previous literature needs to be clarified more. Below, we raise questions and comments, including those related with the above concerns. Only if the authors address all of them sufficiently, I would recommend the manuscript for publication in Nature Communications.

We are grateful for the Referee’s detailed and constructive feedback. We have addressed all their comments and questions below, which resulted in an expanded and significantly strengthened manuscript. We hope that the Referee will now be able to recommend it for publication in *Nature Communications*.

1. *The authors discuss that exponentially many of the energy eigenstates are scarred. They also mention that most of the eigenstates are scarred.*

(a) *While they are mentioned in the manuscript, no clue is discussed in the main text. The authors could briefly explain how they confirm it there.*

We have significantly expanded and clarified the discussion on the quantification of scarring. In particular, we have added a new figure and two new paragraphs to the main text:

“Having seen in Fig. 2 the visual, qualitative features of scarring, we now turn to a quantitative analysis, showing that the eigenstates are anomalously large on the UPOs. We consider the mid-spectrum eigenstates $|E_n\rangle$ of the symmetry sector with zero momentum and left-right mirror parity $+1$, and overlap them with three types of states $|\psi\rangle$: Haar random states, phase-space states $|\{\mathbf{s}_i\}\rangle$, and IS states. To make sure these are treated on par, namely that generic phase-space states are not penalized for being less symmetric than the IS states, we renormalize all the states to have weight 1 on the considered symmetry sector (details

in [43]). That is, we consider the rescaled projection of $|E_n\rangle$ on $|\psi\rangle$, namely $x = \mathcal{D} \frac{|(E_n|\psi)|^2}{\langle\psi|\hat{U}\hat{U}^\dagger|\psi\rangle}$, where \mathcal{D} is the size of the considered symmetry sector and \hat{U}^\dagger the operator projecting on it. Sampling $|E_n\rangle$ and $|\psi\rangle$ yields a probability distribution for x , shown in Fig. 3(a). Haar random states yield the Porter-Thomas distribution e^{-x} [48,49], setting a benchmark for quantum chaotic behaviour. This benchmark is closely followed by the projections of the eigenstates on generic points of the phase space $\{s_i\}$. A deviation from the benchmark is found for projections on the IS states, yielding a fat tail in the distribution of x . This is remarkable: it shows that, due to scarring, the eigenstates can indeed have an anomalously large projection on the UPOs.

To quantify at the same time both aspects of scarring, namely the visual features of Fig. 2 and the fat tail of the projection in Fig. 3(a), we introduce a “scarness” parameter $S = 4\mathcal{D} \times \max_{\text{IS}} \oint Q_\psi$, where $\oint Q$ denotes averaging of Q along each UPO, and \max_{IS} maximization over the IS UPOs (details in [43]). Sampling $|\psi\rangle$ yields a probability distribution for S , shown in Fig. 3(b). While the mid-spectrum eigenstates are often compared to Haar random states, they yield a distribution of the scarness S with a much fatter tail. Indeed, the S of many eigenstates is larger than the S of most Haar states, and the number of scarred eigenstates N_{scarred} grows exponentially with the considered system sizes N (inset).”

FIG. R8. (extract from Fig. 3) **“Quantitative features of scarring.** (a) Distribution of the rescaled overlaps x between mid-spectrum eigenstates and various families of states. The overlaps with Haar random states (red) have distribution e^{-x} (dashed), similarly to the overlaps with generic points $\{s_i\}$ of the phase space (green). By striking contrast, the overlaps with the IS states (blue) yield a much fatter tail, that is, scarring makes some eigenstates anomalously large on the UPOs. (b) The distribution of the scarness parameter S for mid-spectrum eigenstates (blue) has a long tail compared to Haar random states (red). Indeed, the number N_{scarred} of scarred eigenstates, defined as the number of eigenstates with S larger than the 90-th percentile of the Haar states (red dashed line), minus 10% [43], is large and appears to grow exponentially with system size (inset). In (a) and (b) the eigenstates are sampled from the middle of the spectrum, namely among the 10% eigenstates with lowest $|E|$. In (a), the phase-space states are obtained sampling the spins $\{s_i\}$ uniformly and independently from the sphere, and the IS states are obtained sampling s_1 uniformly from the sphere and alternating the other spins at every other site as in Fig. 1(a). Here, $N = 20$ except where otherwise and explicitly specified. All other parameters are as in Fig. 2.”

More details are presented in the revised Supplementary Information, especially on the need for a careful account of the system symmetries when quantifying scarring.

(b) The above claim is discussed in Supplementary Information. There, they count numbers of eigenstates that have larger S [Eq. (S3)] compared with the average S . However, I am not sure if this criterion is appropriate. Even for random wavefunctions, the definition leads to $\chi = 50\%$, which wrongly indicates that exponentially large number of the random wavefunctions are scarred. While χ for energy eigenstates are much larger than 50%, I think the threshold of S to determine the scarred states should be much larger.

We have followed the Referee’s suggestion and changed the definition of χ by increasing the threshold from 50% to 90%, and also subtracted the remaining 10%, so that $\chi = 0$ for Haar random states. The revised analysis, that we report below for the Referee’s convenience, lead to the same conclusion that an exponentially large number of eigenstates is scarred. Quoting from the revised Supplementary Information:

“[...] we consider the N_e mid-spectrum eigenstates, say N'_e the number of them that have S larger than the

90-th percentile of the S for the Haar random states, and introduce the scarring probability χ as

$$\chi = \frac{N'_e}{N_e} - 0.1, \quad (\text{S6})$$

where the -0.1 compensates for the fact that even for random Haar states S is larger than the 90-th percentile 10% of the times (by definition of 90-th percentile). If the eigenstates behaved like Haar random states, one would get $\chi = 0$. But because, instead, many of the eigenstates have S larger than most of the Haar random states, the scarring probability χ can then take relatively large values ($\sim 40\%$ for the considered parameters), indicating that a significant fraction of the eigenstates is scarred by the IS UPOs. Analogously, we can define the number of scarred mid-spectrum eigenstates as

$$N_{\text{scarred}} = N'_e - 10\%N_e = \chi N_e, \quad (\text{S7})$$

which we plotted in the inset of Fig. 3(b) in the main text.”

(c) At least, the statement that most of the eigenstates are scarred seems to be incorrect, since it would indicate $\chi \approx 100\%$ in the thermodynamic limit.

The Referee is right that this statement was incorrect.

Scarred eigenstates are characterized by two key features of the projection Q : (i) qualitatively, it distributes according to the classical trajectories and (ii) quantitatively, it is “anomalously large” on the UPOs. The first feature can be appreciated visually upon plotting Q/Q_{max} and superimposing the classical UPOs: Q distributes in a way that reflects the underlying classical trajectories. Such feature is found by inspection in *most* eigenstates (see Fig. S1, in which 10 *consecutive* eigenstates are shown), and should be contrasted to the case of a random state, that is agnostic of the Hamiltonian and with a distribution Q that bears no information on the classical UPOs (see, e.g., Fig. S2). The second feature, on Q taking an anomalously large value, can instead be appreciated from a careful statistical analysis of Q and its fluctuations across ensembles of eigenstates and random states, as discussed in the previous point and as shown in the new Fig. 3.

This considered, the correct statement is: For *most* eigenstates, the projection Q has a structure reflecting the underlying classical orbits (i); For *many* eigenstates the projection Q is furthermore anomalously large on certain UPOs (ii). The joint effect of both (i) and (ii) is captured by the “scarness” parameter, indicating that exponentially many (but not most) eigenstates are scarred. These aspects are now carefully discussed with the new Fig. 3 (see point above).

2. The authors state that they demonstrate the ubiquity of quantum scarring.

(a) To what extent is it a universal phenomenon? As far as I understand, the authors only consider spin 1/2 system for quantum models (it should be clarified in the main text). What happens if they consider spin 1 systems in the form of (1)? Do they have similar scars?

We thank the Referee for this observation. First of all, we should be clear that we do not claim scarring to be “universal” in a stat-mech sense, but instead ubiquitous, that is, somewhat commonly found across various, popular, and rather general models. The Referee is right that in the main text we only consider $s = 1/2$. In the revised manuscript we specify this explicitly, and refer to the Supplementary Information for data on larger s :

“We shall here focus on the deep-quantum limit of $s = \frac{1}{2}$ (larger s are considered in [43] and yield similar results).”

Indeed, we teamed up with Mr. Long Hei Kwan, who has run simulations for larger s and has thus been added to the author list. With a new section of the Supplementary Information, reported below for the Referee’s convenience, we confirm that spin- s models in the form of (1) indeed yield similar scars.

Larger spins – In the main text we developed a general formalism valid for any spin length s , but tested it focusing on the most quantum case of $s = 1/2$. Here we show that a similar phenomenology of scarring emerges for larger spins $s = 1, 3/2$ and 2 . In Fig. S3 (in analogy with the main Fig. 2) we show that most of the mid-spectrum eigenstates have a projection Q that reflects the structure of the underlying UPOs. In

Fig. S4(a) (in analogy with the main Fig. 4), we show that the structure of the eigenstates underpins memory effects in the time-averaged projection \bar{Q} . Note that, going deeper into the classical limit (i.e., increasing s), the features of the eigenstates and of the time-averaged projection become narrower, in accordance with the decreases of the effective \hbar and in analogy with the single-particle case [11]. In Fig. R10(b,c) we show the scaling with s and N of the return probability, $\bar{Q}(\mathbf{y} \rightarrow \mathbf{y})$, and of the ratio of return and cross probabilities, $\bar{Q}(\mathbf{y} \rightarrow \mathbf{y})/\bar{Q}(\mathbf{y} \rightarrow \boldsymbol{\mu})$. The return probability decays with both N and s , according to the growth of the size of the Hilbert space, $\sim (2s+1)^N$. Yet, for all simulated s the return probabilities remain clearly larger than the cross probabilities, by roughly four times, similar to Fig. 4(c) for spin 1/2 in the main text.”

FIG. R9. (extract from Fig. S3) **Quantum scars for larger spins.** Projection over the manifolds of TI and IS states of 10 consecutive eigenstates taken in the middle of the 0^+ symmetry sector for system size $N = 8$ and spin lengths $s = 1, 3/2$, and 2. Similar to the case of $s = 1/2$, also for larger spins s the eigenstates tend to reflect the structure of the underlying UPOs. All the non-specified parameters are as in the main Fig. 2.

FIG. R10. (extract from Fig. S4) **Weak ergodicity breaking for larger spins.** (a) Time-averaged projection \bar{Q} over the manifold of IS states for $s = 1/2, 1, 3/2, 2$, and system size $N = 8$. The system is initialized on the IS UPOs aligned along \mathbf{y} and $\boldsymbol{\mu}$ in the upper and lower rows, respectively, as marked by the white cross. For all considered s , the time-averaged projection is enhanced on the UPO the system is initialized on, that is, ergodicity is weakly broken. The regions of larger \bar{Q} become increasing concentrated along the UPOs when increasing s . (b) Scaling of the return probability $\bar{Q}(\mathbf{y} \rightarrow \mathbf{y})$ with s and N . (c) Ratio of return probability $\bar{Q}(\mathbf{y} \rightarrow \mathbf{y})$ and cross probability $\bar{Q}(\mathbf{y} \rightarrow \boldsymbol{\mu})$ for various s and N (namely those for which exact diagonalization is amenable – the others are marked as N/A). The initial state is $|\psi_0\rangle = |IS, \mathbf{y}\rangle$ in all cases. The return probability remains around four times larger than the cross probability in all the cases we simulated. Here, we considered the mixed-field Ising model with parameters as in Fig. 2 in the main text.

1(a) - [cont'd] For spin 1 case, we can include different type of interactions in addition to (1), such as the biquadratic interaction. I guess that this interaction considerably affects the presence of the scars since equation (2) may not hold.

This is an interesting point. In the reply above we have only considered increasing s without changing the model in Eq. (1), but indeed a larger s allows extra terms such as $(s_j^z)^2$. We would expect such terms to negatively affect scarring from the TI and IS UPOs of Eq. (3). Yet, it is possible that *new* UPOs would arise, and with them different forms of scarring. For instance, in our previous work [PRL 132, 020401 (2024)] we have considered one such biquadratic terms [Eq. (1)] and still were able to find UPOs and scarring (although in a much more specific model). A detailed study of Hamiltonians beyond our model in Eq. (1) remains a timely question for future research, which we pose in our revised outlook:

“Our work also opens many avenues for theoretical research, posing questions regarding [...], the effect of Hamiltonian terms beyond those in Eq. 1 [...].”

(b) If we consider integrable spin 1/2 systems, such as Heisenberg chain, the scar will not arise, whereas $s \rightarrow \infty$ limit leads to the chaotic behavior. Is that correct?

A few comments are due. Let us first make a general statement, and then directly address the example of the Heisenberg chain.

The idea that some eigenstates have structure along the periodic orbits and that this can leave a mark on the quantum system is general, and can in principle apply to models that are integrable or not, and to periodic orbits that are stable or not. For instance, if the quantum system is integrable, we still expect that some eigenstates will have a large overlap with the periodic orbits, and that this will lead to memory effects as in Fig. 4. So what is so special about the combination “non-integrable + unstable”? That, it is the most surprising. It is somewhat not too surprising to see structure in the eigenstates of an *integrable* system, nor along some *stable* classical orbits. But it is very surprising to see structure for a *non-integrable* model and along *unstable* classical periodic orbits. While the two cases would have lots in common, it thus makes sense to use the word “scar” only for a quantum chaotic system and unstable period orbits. Note that the same applies for single-particle systems: lots of the scar phenomenology can be found along stable periodic orbits and/or in integrable systems, simply one would not invoke the word “scars” in that case.

Let us now turn to the specific example of the Heisenberg chain for spin-1/2 particles, which we have learned is indeed somewhat special from the perspective of scarring. This model is obtained from ours upon setting

$\mathbf{J} = \mathbf{J}\mathbf{I}$ in Eqs. (1-3). The IS orbits consist of precessions around the field $\boldsymbol{\mu}$, that can be generated by the unitary $\hat{U}(\alpha) = \exp\left(-i\frac{\alpha}{2}\frac{\boldsymbol{\mu}}{|\boldsymbol{\mu}|} \cdot \sum_j \hat{\mathbf{s}}_j\right)$. That is, starting from a (product) state $|\text{IS}\rangle$ on an IS orbit, all the others on the same orbit are obtained as $|\alpha\rangle_{\text{IS}} = \hat{U}(\alpha)|\text{IS}\rangle$, for $\alpha \in [0, 2\pi]$. Crucially, this model has a $U(1)$ symmetry, $[\hat{H}, \boldsymbol{\mu} \cdot \sum_j \hat{\mathbf{s}}_j] = 0$, and thus the energy eigenstates also diagonalize $\hat{U}(\alpha)$, namely $\hat{U}(\alpha)|E\rangle = e^{i\beta}|E\rangle$ with $e^{i\beta}$ a phase factor. This directly implies that $|\langle E|\alpha\rangle_{\text{IS}}|^2$ does not depend on α , that is, that the overlap Q of the energy eigenstates $|E\rangle$ on the classical phase space remains *exactly* constant along the IS UPOs. We expect that the general phenomenology would be similar to that discussed in our paper. But because the uniformity of Q along the IS UPOs somehow “piggybacks” on the underlying $U(1)$ symmetry, and because the quantum system is integrable, it appears to us that a sensible nomenclature choice is to *not* use the term scar in this case.

In the revised manuscript, we have commented on these aspects in the following passage:

“We also note in passing that for the Heisenberg model ($\mathbf{J} = \mathbf{J}\mathbf{I}$, not shown here), the projection Q of the eigenstates $|E_n\rangle$ is exactly constant along the IS orbits, owing to the underlying $U(1)$ symmetry $[\hat{H}, \sum_j \boldsymbol{\mu} \cdot \hat{\mathbf{s}}_j] = 0$. By contrast, in the models considered here the structure of the eigenstates on the UPOs does not “piggyback” on any symmetry.”

Several related literature exists, including the works cited in the manuscript.

We thank the Referee for these remarks, that helped us in the revised manuscript to clarify the standing of our work within the existing literature.

(a) *I think that the authors should detail the distinctions and novelty compared with [15,27,30] more.*

[PRL **122**, 040603 (2019)] and [PRX **10**, 011055 (2020)] (namely [27,30] in the old manuscript) focus on the athermal eigenstates of the PXP model. These have been called “scars”, but evidence suggests that they are “regular eigenstates” more than genuine scars. By contrast, we focus on genuine scars in thermal eigenstates. We have clarified by expanding the following passage and adding a few references:

“Growing attention has been recently devoted to certain many-body quantum systems hosting a few eigenstates which violate ETH and are weakly entangled [27-30]. While these have been dubbed “quantum many-body scars”, there is no evidence that they are in fact scars, because they could not be related to UPOs in a chaotic phase space, but rather were often associated to its regular regions [28,31-36]. Such athermal eigenstates are not the focus of our work. Instead, here we consider the thermal eigenstates of many-body systems and show their genuine scarring, due to UPOs in a chaotic phase space.”

In particular, Ref. [15], done by the same authors, already claims that “Here, we find the first quantum many-body scars originating from UPOs of a chaotic phase space.” and “Remarkably, these states verify the eigenstate thermalization hypothesis, and we thus refer to them as thermal quantum many-body scars.” Then, what is the distinctive feature in the present manuscript compared to it?

In our previous work [PRB **110**, 144302 (2024)] (namely [15] in the old manuscript), we have shown one *very specific* example of genuine many-body scarring. The model, phase space, and UPOs were rather contrived (for instance, the considered Hamiltonian was an effective Floquet Hamiltonian, i.e., it was obtained from the Floquet expansion of a driven system, valid for spin-1 particles only). PRB **110**, 144302 (2024) is indeed important because it taught us how to frame the problem of genuine scarring in a many-body system, and showed that it is *possible*. The present work is much broader in scope: it shows that scarring is *ubiquitous*. Our model [Eq. (1)] is very general, including many of the most popular models from condensed matter, and the dynamics [Eq. (2)] and UPOs [Eq. (3)] are very simple. The generality and simplicity are crucial to establish (genuine) scarring as a general concept for many-body systems, making it more accessible for both theorists and experimentalists. In the revised manuscript, we emphasize these points in a new introductory paragraph (see the next point for a quote).

(b) *A similar motivation to connect quantum many-body chaos and unstable periodic orbits is found in, e.g., Refs. [Phys. Rev. Lett. 130, 250402 (2023); Phys. Rev. Lett. 118, 164101 (2017)]. Could you mention the relations with these works?*

We thank the Referee for pointing out these works.

[*PRL* **130**, 250402 (2023)] is highly relevant and we apologize for having overlooked it. We very much align with their perspective on genuine scarring, namely scarring as in billiards. They consider a specific example of interacting bosons in a ring lattice and emphasize the few-body semiclassical regime of many bosons in few sites (while also contemplating the deep-quantum regime for larger systems). In the revised manuscript, we cite this work prominently in the following “gap-opening” passage:

“Scarring has long been known within single-particle quantum chaos, but its generalization to many-body quantum systems, whose study requires modern quantum simulators and more advanced numerical tools, has remained virtually unexplored, limited to specific recent instances of interacting bosons in a ring lattice [15] and a periodically driven spin-1 chain [16].”

The work above indeed provides further motivation for ours. Quoting them, “In the context of the widely considered spin-chain-like systems such a study is, however, hampered by the fact that those quantum Hamiltonians do not have an obvious classical counterpart”, a gap which we filled. In general, the point is indeed that only a couple of works have considered genuine scarring in many-body systems, and only in very specific situations. Our work has a much broader scope, showing that scarring is ubiquitous, naturally appearing in many of the most popular spin chain models, and thus establishing it as a very general concept.

[*PRL* **118**, 164101 (2017)] considers the link between the spectral properties of a quantum many-body system and the underlying classical periodic orbits. While their mission is broadly speaking aligned with ours, the main difference is that they do not discuss scarring.

Another interesting parallel between our work and the two above is that also in their case $N = 4k$ is special: for [*PRL* **130**, 250402 (2023)] it leads to “hopping suppressing” configurations, in analogy with our “interaction suppressing” states; for [*PRL* **118**, 164101 (2017)] it yields some special manifolds of periodic orbits (although not of IS type) with a strong impact on the spectral properties of the quantum system.

We better contextualize our work in the following new paragraph of the Introduction, in which the works above (and two more) are cited:

“Beyond [15,16], note that such genuine scarring has also been shown in the Dicke model [37] and for a spinor condensate [19] (also observed in experiments [38]), that with their all-to-all interactions sit somewhere in between few- and many-body systems. Moreover, a semiclassical analysis of the periodic orbits of a quantum many-body system was presented for a periodically driven spin chain in [39], although not in relation to scarring.”

Moreover, we added a footnote to highlight the analogy between the IS UPOs and the periodic orbits observed in the works above:

“System sizes $N = 4k$ led to special effects also in previous related work: in [39] they resulted in special manifolds of periodic orbits with enhanced impact on the spectral properties of a periodically driven spin chain, whereas in [15] they were related to “hopping suppressing” UPOs of bosons in a lattice, in analogy with our IS UPOs.”

Finally, our standing in the literature is further clarified by the revised and more concrete title:

“Genuine quantum scars in many-body spin systems”.

(c) In the famous paper by Srednicki [Phys. Rev. E 50, 888 (1994)], he also discussed the effect of genuine scars in considering quantum thermalization. I think the authors could mention this.

We thank the Referee for this illuminating remark. Srednicki shows that, under Berry’s conjecture, the eigenstates predict a thermal distribution for the momenta of a particle in a gas, that is, that the eigenstates fulfill ETH. Quantum scars are a violation of Berry’s conjecture, and so question whether ETH holds. Srednicki comments: “Scars represent violations of Berry’s conjecture which are quite obvious when one looks at the Wigner density of an energy eigenstate in phase space, since there the scars appear with a “signal-to-noise” ratio of 1:1. Once we integrate out all of the coordinates and most of the momenta, however, the scars fade away almost completely.”

Srednicki's work is thus help to contextualize our claims. Scars can be clearly visible at the level of wavefunction projection on phase space (either for eigenstates, in Fig. 2, or for the time averaged dynamics, in Fig. 4). Such projections are many-body observables and can thus evade ETH. For simpler few-point observables (like the magnetization or a two-point correlator), instead, computing the expectation value effectively involves an integration over the whole phase space (as can be made precise within the Wigner formalism), which washes away the effect of scars and restores thermalization.

In the revised manuscript we have added the following explicit reference to Srednicki's work:

“Indeed, as long noted by Srednicki [51], in many-body systems the effect of scarring is mostly washed away when integrating over the phase space, as effectively done when computing the expectation value of simple observables (e.g., $\langle s_j^z \rangle$).”

4. I have several questions about Q .

(a) Some additional explanations may be needed about how Q is justified as the signal of quantum scars. As mentioned by the authors, they only consider projections along UPOs in the two-dimensional manifold, although the dimension of the entire phase space is much larger (increasing with N). Is there an explanation of why this projection does not cause a problem in discussing scars? For example, for original scars in quantum billiards, such projections with large dimensional reductions are not usually employed.

We thank the Referee for this remark. The high dimensionality of the phase space is merely a result of piecing together many single-particle phase spaces, which is a peculiarity of many-body systems. Scarring was traditionally studied in single particle systems, like billiards, in which the phase space is low-dimensional. But the theory of scarring does not make any specific assumption on the dimensionality of the phase space, and indeed, as the Referee pointed out in the previous point, the possibility of scarring in many-body systems (thus, in highly dimensional phase spaces) was already contemplated by Srednicki in 1994. Thus, while the high dimensionality requires an extra step with abstraction and visualization, it is very natural and per-se not problematic.

Moreover, in the paper by Heller (1984), the inner product between the eigenfunction and gaussian wave packets are considered, while this paper discusses the overlap with the classical states whose spins are maximally polarized. Do you have comments on these differences?

The fully-polarized states that we consider are indeed close analogues of the Gaussian wavepackets considered for billiards, but for spins.

Think of just one spin for simplicity. Intuitively, full polarization implies that the quantum spin $|\mathbf{s}_j\rangle$ tries to be somehow as close as possible to the classical spin \mathbf{s}_j , up to the unavoidable quantum fluctuations in the directions transverse to \mathbf{s}_j . In phase space, the spins can be described by a Wigner distribution W on the Bloch sphere, which has a pretty convoluted expression [PRA 49, 4101 (1994)]. Because the phase space is the surface of a sphere, the Wigner function cannot be exactly a Gaussian. And yet it almost is, and indeed for large s it can be shown that $W \approx (\pi s)^{-1} \exp(-s(s_x^2 + s_y^2))$, that is, the Wigner function is a Gaussian localized around $\mathbf{s} = \mathbf{z}$ with standard deviation $\sqrt{2/s}$ in the transverse directions [Ann. Phys. 325, 1790-1852 (2010)]. Because computing the Wigner distribution for spins can be challenging, we have instead focused on a Husimi-type distribution Q , that is morally analogous to the Wigner distribution, but much easier to compute. And indeed, also for Q we can see that the states $|\{\mathbf{s}_j\}\rangle$ have a pseudo-Gaussian distribution, as we now clarify in a footnote:

“For instance, the projection of a state $|\{\mathbf{s}'_j\}\rangle$ itself on a point $\{\mathbf{s}_j\}$ of the phase space reads $Q = \prod_j \left(\frac{1 + \mathbf{s}'_j \cdot \mathbf{s}_j}{2} \right)^{2s} \approx \exp\left(-\frac{s}{2} \sum_j \theta_j^2\right)$, with θ_j the angle between \mathbf{s}_j and \mathbf{s}'_j and where the approximation becomes more accurate when increasing s . Indeed, the states $|\{\mathbf{s}_j\}\rangle$ are the natural generalization of Gaussian states for spins.”

(b) While the authors mention that the high dimensionality of the classical phase space makes visualizing Q generally complicated, I think that it is not easy to compute, either, since we need to sample exponentially large numbers of points to obtain Q for many-body systems.

Performing a full tomography of a many-body wavefunction, namely computing $Q(\{\mathbf{s}_j\})$ on a fine mesh that covers the entire phase space (thus, with $\sim e^{\mathcal{O}(N)}$ points) would indeed be challenging. But this is not necessary for any of our results. For the sake of clarity, let us emphasize that, when we consider Q on the manifolds of UPOs, we are *not* integrating out the rest of the phase space, on which no computation is thus needed. To appreciate scarring one must compute Q on just a few points of interest in the many-body phase space. Indeed, in the extreme case, for instance in a minimal-resource experiment, it would be enough to consider just two points, as we elaborate in our reply to point 6 below.

(c) *The scaling of Q is conjectured to scale as $Ne^{-\mathcal{O}(N)}$. I think that this scaling is important to understand the consistency with thermalization. However, no clue seems to be found for this statement.*

We agree that it is worth explicitly motivating this scaling. It follows from symmetry arguments, as we now specify in the following footnote:

“The Hilbert space has size $\sim e^{\mathcal{O}(N)}$. Due to translational symmetry, it is divided in N momentum sectors, each of size $\sim e^{\mathcal{O}(N)}/N$. The average overlap scales as the inverse sector size, thus as $\sim Ne^{-\mathcal{O}(N)}$.”

(d) *Furthermore, I feel that it is more direct to consider the scaling with s if one wants to compare with quantum scars in semiclassical systems. Namely, the role of \hbar in semiclassical systems is played by $1/s$ in the present manuscript, rather than $1/N$. So, I wonder what is the s dependence of Q .*

Our entire model and framework extend to larger s , and in the revised Supplemental Material we now show similar scarring results for $s = 1, 3/2$, and 2 (see reply to point 2 above). We agree with the Referee that the semiclassical limit $s \rightarrow \infty$ creates a direct connection to billiard-like scars. Yet, this is precisely why in the main of our paper we prefer to focus on the case of $s = \frac{1}{2}$ instead. Let us elaborate.

Consider a chaotic Bunimovich billiard. The ground and first excited states, shown in Fig. R11, appear rather regular, similar to those of a rectangular box potential – chaos is not really allowed to manifest at such low energies. More interesting are the highly excited states (namely states in the semiclassical regime). For them, chaos is a priori allowed (indeed expected), and this is why scarring, namely a form of suppression of chaos, is so surprising and special. Semiclassicality can be seen as a handle to enable chaos, bringing us into a regime where seeing structure in the form of scars is exciting.

FIG. R11. **Semiclassicality as a handle to chaos in a billiard.** The wavefunction $|\psi_n(x, y)|^2$ of a few eigenstates of the Bunimovich billiard is shown. The ground and first five excited states ($n = 1, 2, \dots, 6$) appear rather regular. Chaos can manifest only at larger energies, i.e., in the semiclassical regime (e.g., for $n = 5056$) and be to some extent suppressed by quantum scarring (e.g., for $n = 5058$).

Many-body systems introduce a second handle to reach a quantum chaotic regime, namely the system size N . At large enough N , the system becomes quantum chaotic even if far away from the classical limit ($s = 1/2$ for us). This is the fundamental difference distinguishing many-body and single-body scarring, which explains our choice to focus on $s = 1/2$ in this paper, and which also replies to Referee’s question 5(a) later. We now emphasize this point in the following passage:

“In single-particle systems, chaos—and consequently scarring—become meaningful only in the semiclassical regime, such as at sufficiently high energies in quantum billiards [11]. By contrast, many-body systems can achieve a quantum chaotic regime through sufficiently large system sizes N . This distinction enables a fundamentally new type of scarring unique to many-body systems. For instance, we found that the $s \rightarrow \infty$ classical dynamics in Eq. (2) scars the quantum system all the way down to the deep-quantum limit of $s = \frac{1}{2}$.”

(e) *In Supplementary Information IV), they consider $s = 1/2$ in Eq. (S36) even for classical systems. How is this choice justified?*

The short answer is that the results would be analogous for any s , but $s = 1/2$ makes numerics easier.

Consider a classical probability distribution $\rho(0)$ that is initially well localized around a point of the phase space. The dynamics of the probability distribution $\rho(t)$ is governed by the Liouville's equation. The point we want to make in the paper is that the time average $\bar{\rho}$ of $\rho(t)$ is independent of the initial condition, that is, that the system is classically ergodic.

One way to make this point would be to brute-force simulate the Liouville's equation, compute $\bar{\rho}$, and verify it does not depend on the initial condition. This is computationally extremely challenging. A simpler approach, à la Monte Carlo, is to consider an ensemble of neighboring initial conditions $\{\mathbf{s}_j^{(r)}(0)\}$, with $r = 1, 2, \dots, R$, evolve them over a long time to get trajectories $\{\mathbf{s}_j^{(r)}(t)\}$, and obtain a discretized $\bar{\rho}$ by making a histogram in phase space. Specifically, to know $\bar{\rho}(\{\mathbf{s}_j\})$ at the point $\{\mathbf{s}_j\}$ of the phase space one would have to construct a bin around $\{\mathbf{s}_j\}$, and count how many trajectories are present in it, on average in time. Because the phase space has a non-trivial structure (product of N spheres), there is some ambiguity in how exactly to construct such bin.

To lift this ambiguity, and to make numerics somewhat smoother, we prefer to say that all the trajectories contribute to $\bar{\rho}(\{\mathbf{s}_j\})$, but in a way that depends on their distance from the query point $\{\mathbf{s}_j\}$: the larger the distance, the smaller the contribution. The overlap $\left| \langle \{\mathbf{s}_j\} | \{\mathbf{s}_j^{(r)}\} \right|^2 = \prod_j \left(\frac{1 + \mathbf{s}_j \cdot \mathbf{s}_j^{(r)}(t)}{2} \right)^{2s}$ comes to help: it naturally weighs how far $\{\mathbf{s}_j^{(r)}(t)\}$ is from $\{\mathbf{s}_j\}$. This is indeed the spirit of \bar{Q}_c in our manuscript, that plays the role of $\bar{\rho}$ here. The overlap contains the exponent $2s$, which decides how quickly the contribution should decay with the distance. But note: s here does not have a particular physical meaning, it should be regarded as just an arbitrary parameter in the definition of the distance (e.g., it does not need to be half-integer – choosing $s = \sqrt{2}/\pi$ would be ok).

The larger s , the more heavily a misalignment between $\mathbf{s}_j^{(r)}$ and \mathbf{s}_j is penalized. In this sense, we can say that s controls the size of some effective phase-space bin. The volume of the effective bin decreases exponentially with s : if s is large, it becomes very unlikely that a trajectory will meaningfully contribute to $\bar{\rho}$, and the Monte Carlo sampling would require a very large number $R \sim e^s$ of trajectories to converge. This is just a practical issue: $s = 1/2$ allows convergence “already” for $R = 2 \times 10^5$ (which still takes ~ 1 day to run). Simulations for a larger s are expected to yield the same result (independence of $\bar{\rho}$ on the initial condition), but would require much longer to run, because the required R would be much larger (e.g., we can estimate $R \sim 10^{10}$ for $s = 1$).

We have clarified this point with the following addition to the Supplementary Information:

“The parameter s controls how heavily a misalignment between $\mathbf{s}_j^{(r)}$ and \mathbf{s}_j is penalized. The larger s , the more unlikely that a trajectory will meaningfully contribute to the sum, and the larger the number of samples R required for convergence. In Fig. 3 we consider $s = 1/2$, for which a good convergence is obtained for $R = 2 \times 10^5$ samples. This choice is ultimately arbitrary, and we expect that the same key finding (that \bar{Q}_c does not depend on the initial condition) would be found for larger values of s , although at the cost of a larger computational burden.”

5. The authors stress that the scars are found in many-body systems.

(a) Is there any unique feature of quantum scars in many-body systems, compared with those in few-body systems (such as quantum billiards)?

Yes, there is. For a full reply, we refer point 4d above. Here, we just report again the relevant passage from the revised manuscript:

“In single-particle systems, chaos—and consequently scarring—become meaningful only in the semiclassical regime, such as at sufficiently high energies in quantum billiards [11]. By contrast, many-body systems can achieve a quantum chaotic regime through sufficiently large system sizes N . This distinction enables a fundamentally new type of scarring unique to many-body systems. For instance, we found that the $s \rightarrow \infty$ classical dynamics in Eq. (2) scars the quantum system all the way down to the deep-quantum limit of

$s = \frac{1}{2}$.”

(b) *In the introduction, the authors state that their quantum scars yield nontrivial effects in quantum many-body systems, mentioning some examples, such as integrability, many-body localization, etc. However, this comparison looks a bit peculiar because those examples all break the eigenstate thermalization hypothesis, whereas quantum scars presented in this manuscript do not.*

Exactly: the list only contain ETH-breaking mechanisms, which is why we are so excited to have found one in which ETH is fulfilled instead! That’s precisely the point we want to make with that list: it is hard to think of nontrivial effects in *generic* nonequilibrium quantum many-body systems, without relying on some form of emergent integrability or ETH breaking. The divide between the list and our work is thus a point of strength in our motivation, as we better emphasize with the following slight revision:

“Our work adds quantum scarring to the (short) list of mechanisms yielding nontrivial effects in many-body quantum systems out of equilibrium, such as integrability, many-body localization, Hilbert space fragmentation, and non-thermal eigenstates in an otherwise chaotic spectrum. All these rely on an explicit or emergent partial integrability and host ETH-breaking eigenstates. By contrast, scarring establishes a deviation from chaos in the thermal eigenstates of generic non-integrable many-body systems, where one would least expect it.”

6. The authors claim that their predictions are straightforwardly verified in experiments. However, as mentioned in the manuscript, the effect decays exponentially as N is increased. Therefore, it would be hard to detect as we consider many-body limit. The authors should comment on this apparent problem.

We thank the Referee for this question, that motivated us to clarify and provide more context in the revised manuscript. The same question was asked by Reviewer 1 in their point (6), and so we reply in the same way.

First of all, directly measuring the many-body eigenstates is of course not realistically possible, and scarring would thus have to be indirectly measured via its effect on the dynamics, as in Fig. 4. A few considerations are due:

(i) The full “tomography” of Fig. 4, in which θ and ϕ are used to scan the manifold of UPOs, would not be necessary. To prove the point that the system remembers its initial condition, namely that \bar{Q} is larger on the orbit on which the system was initialized, it would be enough to consider one initial condition and the projection on two points in phase space (one on the orbit, \bar{Q}_{on} , and one not, \bar{Q}_{off}), for a total of just 2 evaluations of \bar{Q} ;

(ii) The projection \bar{Q} would be measured from repeated experiments. Because \bar{Q} is exponentially small in N , the number of repetitions is exponentially large. However, there is no need to push N to be as large as possible, and a meaningful experiment to prove scarring could already be done for $N = 12$, for which $\bar{Q}_{\text{max}} \sim 10^{-2}$ and $10/\bar{Q}_{\text{max}} \sim 10^3$ repetitions should suffice, which is reasonable;

(iii) Measuring a local observable $\langle \hat{O} \rangle = 0.1$ with precision 10^{-3} would be hard due to noise thresholds. However, while the *absolute* value of \bar{Q} is small, its *relative* fluctuations are large [e.g., $\bar{Q}_{\text{on}}/\bar{Q}_{\text{off}} \sim 4$ in Fig. 4(c)]. Thus, even if noise and imperfections would introduce a large uncertainty in \bar{Q} (say, pessimistically, 50%), the contrast between \bar{Q}_{on} and \bar{Q}_{off} , hence scarring, would still be detected;

(iv) Indeed, and most importantly, similar measurements have already been done! Something that we only recently came to appreciate is that Google’s famous quantum supremacy experiment [Arute *et al.*, *Nature* **574**, 505–510 (2019)] deals with essentially the same situation: they initialize a many-body wavefunction, time evolve it, project it on bitstrings, average over many runs, compute the (exponentially small) bitstring probabilities, and argue for genuine quantum features in it. Indeed, our $Q(\theta, \phi)$ is nothing but a bitstring probability, just with respect to a rotated basis, which can be obtained by applying suitable and simple one-bit unitaries right before measuring. It should thus be possible to verify our findings in ever improving state of the art quantum simulators and computers, with a protocol in very close analogy with Google’s from six years ago.

In the following passage of the revised manuscript we now better contextualize and motivate our claim on the feasibility of an experiment:

“In particular, by repeatedly preparing a product state, letting it evolve, and measuring it in a different basis, one should be able to show that the system is more likely to be found on the UPO it was prepared on (Fig. 4), which is a direct consequence of quantum scarring. This protocol is in very close analogy with what has been done in [49,57], in which the many-body bitstring probabilities – essentially the same as our projection Q – have been measured to detect genuine quantum effects.”.

7. In calculating (S3), is it easy to take the maximum over all UPOs?

Yes, it is relatively easy. The computational bottleneck is finding the eigenstates from exact diagonalization. Once that is done, it is then relatively straightforward and quick to generate many product states, project the eigenstates on them, and perform the statistical analysis of Q across phase space (including the brute-force computation of the maximum over all UPOs).

8. After (S32), they find $6N$ eigenvalues, although the original problem (S21) will have $3N$ eigenvalues. How do they reconcile with each other?

The Referee is right that this is an inconsistency. The truth is that there are $3N$ eigenvalues, N of which vanish. The expression of the eigenvalues in Eq. (S32) was correct, but one should be careful with the multiplicities. We have expanded this section of the Supplementary Information to clarify. We report the relevant part below, following the equation numbering of the revised manuscript.

“The eigenproblem reads

$$\frac{d\epsilon_k}{dt} = \lambda\epsilon_k = -(1+i)\sin(k)\mathbf{M}\epsilon_{k-\pi/2} + (1-i)\sin(k)\mathbf{M}\epsilon_{k+\pi/2}, \quad (\text{S32})$$

that is,

$$\lambda \begin{pmatrix} \epsilon_k \\ \epsilon_{k-\pi/2} \\ \epsilon_{k+\pi/2} \\ \epsilon_{k+\pi} \end{pmatrix} = \begin{pmatrix} 0 & -(1+i)\sin(k)\mathbf{M} & (1-i)\sin(k)\mathbf{M} & 0 \\ (1-i)\cos(k)\mathbf{M} & 0 & 0 & -(1+i)\cos(k)\mathbf{M} \\ (1+i)\cos(k)\mathbf{M} & 0 & 0 & (i-1)\cos(k)\mathbf{M} \\ 0 & (i-1)\sin(k)\mathbf{M} & (1+i)\sin(k)\mathbf{M} & 0 \end{pmatrix} \begin{pmatrix} \epsilon_k \\ \epsilon_{k-\pi/2} \\ \epsilon_{k+\pi/2} \\ \epsilon_{k+\pi} \end{pmatrix}. \quad (\text{S33})$$

That is, we have obtained $N/4$ sets of 12-dimensional eigenproblems, for a total of $3N$ eigenvalues. Each eigenproblem is associated to a set of four momenta, namely $k, k - \frac{\pi}{2}, k + \frac{\pi}{2}$, and $k + \pi$. To label each set we can consider values of k up to $\frac{\pi}{2}$, namely $k = \frac{2\pi}{N} \times (1, 2, \dots, \frac{N}{4})$, to which we will henceforth restrict. With the assistance of a computer, we find that the 12 eigenvalues of Eq. (S33) are $(0, 0, 0, 0, ae^{+i\frac{\pi}{4}}, ae^{+i\frac{\pi}{4}}, ae^{-i\frac{\pi}{4}}, ae^{-i\frac{\pi}{4}}, ae^{+i\frac{3\pi}{4}}, ae^{+i\frac{3\pi}{4}}, ae^{-i\frac{3\pi}{4}}, ae^{-i\frac{3\pi}{4}})$, where a is some real positive number. To find it, we iterate Eq. (S32) once to get

$$\lambda^2\epsilon_k = -(1+i)\sin(k)\cos(k)\mathbf{M}^2(-(1+i)\epsilon_{k-\pi} + (1-i)\epsilon_k) \quad (\text{S34})$$

$$\begin{aligned} & - (1-i)\sin(k)\cos(k)\mathbf{M}^2(-(1+i)\epsilon_k + (1-i)\epsilon_{k+\pi}), \\ & = 2i\mathbf{M}^2\sin(2k)\epsilon_{k+\pi}, \end{aligned} \quad (\text{S35})$$

which we iterate again, to close the equation, getting

$$\lambda^4\epsilon_k = -4\mathbf{M}^4\sin^2(2k)\epsilon_k. \quad (\text{S36})$$

Again with the help of a computer, we find that \mathbf{M}^2 has one vanishing eigenvalue and two degenerate eigenvalues $\alpha = -\mathbf{s}[\text{adj}(\bar{\mathbf{J}})]\mathbf{s}$, thus getting $a = \sqrt{2|\alpha\sin(2k)|}$.

Putting all the results together, we thus find that the $3N$ eigenvalues of the problem are 0, with multiplicity N , and

$$\lambda_{k,m} = \sqrt{2|\alpha\sin(2k)|}e^{i\frac{\pi}{4}+im\frac{\pi}{2}}, \quad (\text{S37})$$

with multiplicity 2 and for $m = 1, 2, 3, 4$ and for $k = \frac{2\pi}{N} \times (1, 2, \dots, \frac{N}{4})$. Note: the fact that N of the $3N$ eigenvalues vanish is simply due to the fact that the three components of the spins are not independent, but constrained by $|\mathbf{s}_i|^2 = 1$. ”

9. In (S34), λ does not depend on N . Is it because the authors consider perturbations around specific UPOs (i.e., IS UPOs)?

Yes, λ does not depend on N because the perturbations are specifically around the IS UPOs. The non-dependence on N can be appreciated from our mathematical derivation: while Eq. (2) for the spin dynamics involves $2N$ degrees of freedom $\{\theta_j, \phi_j\}$, and could in principle lead to an N -dependence of the eigenvalues, the linear stability analysis around the IS UPOs leads to a decoupling of the momenta, which appear in groups of four in Eq. (S33), see the reply to the previous question, excluding a dependence of λ on N .

To please the Referee's and our curiosity, in the following Figure R12 we present some numerics in which we compute the Lyapunov exponent in other points of the phase space (that is, away from the IS UPO). To do this, for each system size N we consider $R = 100$ points sampled uniformly across the phase space, namely with each spin sampled independently and uniformly on the surface of a sphere. The R initial conditions are evolved up to a time $t = 5$ by integrating the classical dynamics in Eq. 2. By considering all the possible perturbations of each initial condition, for each trajectory we compute the monodromy matrix $M^{(r)}$, diagonalize it, and obtain the most unstable Lyapunov exponent $\lambda^{(r)}$, with $r = 1, 2, \dots, R$ labeling the trajectory. We then plot the average of $\lambda^{(r)}$, using the standard deviation as errorbar (average and standard deviation are computed with respect to r , that is, to the ensemble). While making definite and precise statements would require more advanced numerics, these preliminary results suggest that the maximum Lyapunov exponent λ does indeed grow with N (possibly as $\sim \log N$) for generic points of the phase space.

FIG. R12. **Dependence of the maximum Lyapunov exponent λ on the system size N .** The maximum Lyapunov exponent associated to the IS UPOs does not depend on system size (blue, and as expected from Eq. S38). Instead, that associated to the trajectories starting from random points of the phase space appears to grow with N , confirming the intuition from the Referee. Here, we have considered the Ising model with $J_{zz} = -4$. The other unspecified parameters are as in Fig. 1c.

We have commented on this point in the following passage: “Note, the fact that λ in Eq. S38 does not depend on N is not obvious a priori: it is a special feature of the considered IS UPOs, for which the dynamics decouples for groups of just four momenta in Eq. S33.”

10. There are some typos.

(a) On page 2, “scarring can be expected when $\lambda\omega < 1$ ” \rightarrow “scarring can be expected when $\lambda/\omega < 1$ ”

(b) In (S1) and (S2), some $\sigma_j^z \sigma_{j+1}^z$ should be $\sigma_j^x \sigma_{j+1}^x$ or $\sigma_j^y \sigma_{j+1}^y$.

(c) In Fig. S1, 4181 and 4182 would be 4182 and 4183, respectively.

We have now fixed the typos.

In conclusion, we want to genuinely thank the Referee for their very detailed, insightful, and constructive criticism, that helped us to expand and significantly strengthen our manuscript. We hope the Referee will now recommend our paper for publication in *Nature Communications*.

Reviewer 4

The Reviewer has co-reviewed the manuscript with one of the reviewers who provided the listed reports.

Reply to the Referees – 2nd Round

Note. This reply is followed by a brief list of changes. In this reply, when we quote the manuscript we mark the changes in light blue font.

Reviewer 1 and 2

This is a second report on “Genuine quantum scars in many-body spin systems” by Pizzi et al. First off, I want to express appreciation to the authors for taking the time and effort to reply very thoroughly to both sets of Referee reports. I have thoroughly scrutinized the authors’ reply to my first report.

I am glad that they have taken my suggestion and performed a quantitative analysis of the degree of scarring claimed as opposed to relying on an arguably loose and not too meaningful criteria as was done in the first version of the manuscript. The main new changes are encapsulated in Fig 3 and Fig 4.

We thank the Referee for engaging in depth with our manuscript and replies and for providing valuable feedback and suggestions. We are glad that they have appreciated the more quantitative analysis in our revised manuscript.

Indeed, I agree with the authors that there is a clear quantitative difference (Fig 3a) between the distribution of (rescaled) overlaps between mid-spectrum eigenstates and Haar random states (and phase space states) versus IS states. I find the deviation from the Porter-Thomas deviation interesting.

Fig 3b shows also a difference of the distribution of the scarness parameter for mid-spectrum states compared to Haar random states, showing that the mid-spectrum states do not behave like Haar random states. I too find this interesting.

The net take-home message is that there is enhancement of overlaps of quantum many-body eigenstates/dynamics on the UPOs of the classical model.

We are glad that the Referee found interesting our new Fig. 3 showing the enhancement of the overlaps of quantum many-body eigenstates and dynamics on the UPOs of the classical model.

At the same time, as the authors themselves have pointed out, this effect is very weak, in the sense it goes down exponentially (as expected) with system size, so the enhancement is only relative (indeed supported by Fig 4).

The fact that the effect is weak is important experimentally: while I agree with the authors that bit-string probabilities can be measured in modern day quantum simulators nowadays, the exponentially small enhancement they are touting therefore needs an exponentially large number of measurements, limiting the observation of their effect to very small system sizes.

The absolute wavefunction amplitudes are exponentially small, but the relative enhancement is not. In this sense, we would not talk about a “very weak effect”, but about a large effect on a small quantity (see also the reply to point 1 below). We agree that, in an experiment, N should not be too large (e.g., $N = 16$ or 20). An upper limit to N would be frustrating if trying to make a claim on $N \rightarrow \infty$, as important, e.g., for a phase transition. But “ $N \rightarrow \infty$ ” is not part of our claim, and limiting N is no problem. As argued in our first report, this is in close analogy with semiclassical systems: to appreciate genuine scarring one should take \hbar^{-1} and 2^N large enough to enable chaos, not as large as possible.

Thus, while I appreciate the message of the authors that there is a weak effect on the structure of many eigenstates, which is opposed to the standard way that quantum many-body scars have been discussed in the literature — that of a very strong effect (athermal entanglement, say) but for very few eigenstates, I am on the fence of whether this constitutes a novel enough result for a journal like Nat Comms. After all, there is no contradiction with the statements of ETH on quantum thermalization of local

observables and the rate at which this equilibrium is achieved with system size. Note again I am not saying there are no new results in the paper: it is certainly curious to me the authors' observation of the enhancement on the UPOs — and I think this result should be published in some form — I am just unsure of the profoundness of this in the grand scheme of things. Perhaps the editors can seek another opinion.

We thank the Referee for their remark and we are glad they appreciate our message. Regarding novelty, the Referee appears to some extent disappointed that ETH is not violated (“after all, there is no contradiction with [...] ETH”). Our perspective on this point is: while breaking of ETH makes PXP-like scars remarkable, it is not distinctive of genuine scarring and should not be used as a metric to judge our work. We are not racing the PXP model on ETH breaking, and the question we ask is fundamentally different. Namely, can we find beyond-random-matrix features in the many-body wavefunction, *in spite of ETH being fulfilled?*

First, we should echo our reply to the first round of reports, namely, building a parallel between our work and single-particle scarring. In single-particle scarring, the size of the accessible Hilbert space \mathcal{H}_{acc} increases with $\hbar \rightarrow 0$, the weight of the eigenstates on the UPOs decays with \mathcal{H}_{acc} , and $\langle E|\mathcal{O}|E\rangle$ approaches the thermal value for “reasonable” observables (e.g., in a billiard, \hat{x}^2 and $\hat{x}^2\hat{y}^2$, but not $|x, y\rangle\langle x, y|$). In our model, the size of the Hilbert space \mathcal{H} increases with N , the weight of the eigenstates on the UPOs decays with \mathcal{H} , and $\langle E|\mathcal{O}|E\rangle$ approaches the thermal value for “reasonable” observables (e.g., \hat{s}_1^x and $\hat{s}_1^x\hat{s}_2^x$, but not $|\{\mathbf{s}_j\rangle\rangle\langle\{\mathbf{s}_j\rangle|$). In both cases, scarring is not about ETH breaking, but about the wavefunction amplitudes in the chaotic regime (that is, at large but finite \hbar^{-1} or N).

Indeed, even if admittedly small, the wavefunction amplitudes are quickly becoming a central object of theoretical and experimental investigation of many-body quantum chaos, see for instance [*Nature Physics* **14**, 595 (2018); *Nature* **574**, 505 (2019); *Physical Review X* **14**, 041051 (2024); *Nature* **638**, 79–85 (2025); *Physical Review Letters* **134**, 010401 (2025); *Physical Review Letters* **134**, 050405 (2025); *arXiv* 2403.11971 (2024); *arXiv* 2404.10057 (2024)]. These high-profile works are not put off by thermalization, nor by the projections being exponentially small. Rather, the projections are embraced for what they are, and valued as a new facet of many-body quantum chaos, subject to universal behavior (Porter-Thomas distribution) and in which genuinely quantum effects can be found (most notably for Google’s quantum supremacy). Within this context, our work unveils new features of such projections, namely the reflection of the classical orbits and deviation from the Porter-Thomas distribution. This is in spite of chaos, drawing a clear connection to billiard-like quantum scars, and proving a measurable and genuinely quantum memory effect. From this perspective, our findings appear timely and of fundamental importance for the understanding of scarring and many-body quantum chaos more broadly, making *Nature Communications* a suitable publication venue.

We have extended the following passage to emphasize this perspective, adding the works above to the bibliography:

“[...] the projection of a wavefunction $|\psi\rangle$ on the classical phase space then reads $Q = |\langle\{\mathbf{s}_i\}|\psi\rangle|^2$ [47]. Being increasingly accessible in quantum computers and simulators, where for \$s = 1/2\$ they are often called bitstring probabilities, such projections are quickly emerging as a key object of investigation in many-body quantum chaos [48-55].”

Moreover, we comment in the Conclusion:

“This protocol is in very close analogy with what has been done in [48,49], in which the many-body bitstring probabilities – essentially the same as our projection Q – have been measured to detect genuine quantum effects. Indeed, accessing complex quantities not described by ETH, such as the bitstring probabilities, allows to search for new physics in spite of the thermalization of local observables, as we have proven for scarring.”

Additional comments:

1. Before any form of publication, I would urge the authors to state clearly in the abstract that the enlarged weight they are pointing out is a very weak effect and is not contradictory to statements of ETH.

We are clear in the abstract that the eigenstates are “thermal and strongly entangled”, and we explicitly emphasize in the text that the wavefunction intensities are exponentially small. We believe that the wording

“a very weak effect” would read as downgrading our findings in an unfair and unjustified way. This is because of similar reasons as above. Our findings might be seen as a very weak correction to local observables and ETH, but this is not the right perspective to judge our work. We, as many others recently, focus on the wavefunction amplitudes, and thus we do not find a “very weak effect”, but a large effect on a small quantity (e.g., the $\sim 400\%$ enhancement in Fig. 4), which reads very different.

We agree with the Referee that mentioning explicitly ETH in the abstract can be helpful. We have thus extended: “[...] even at long times and despite the system being fully thermal and the eigenstate thermalization hypothesis fulfilled.”

2. Also, in the inset of Fig 3b (and also the abstract), the authors state that the number of scarred eigenstates is ‘large’ and ‘appears to grow exponentially with system size’. However, again like in my first report — I caution the authors against such hyperbolic statements: ‘large’ should always be in relation to the growth of some reference function. In this case, one should overlay the function $c2N$ on the inset, since $2N$ is the dimension of the full system size. I have plotted this on my computer. What I see is that the number of “scarred eigenstates”, according to their criteria of scariness, seems a fixed ratio of the total number of eigenstates.

We thank the Referee for this remark. We have reworded to improve concreteness and clarity: from “is large and appears to grow exponentially with system size” to “appears exponentially large in system size”. We can indeed confirm that this statement is precise, as we now show more explicitly.

The number of scarred eigenstates M_{scarred} only accounts for the eigenstates among the 10% of the 0^+ symmetry sector. Thus, the base against which M_{scarred} should be compared is M_e , namely the number of mid-spectrum eigenstates in 0^+ . The ratio between M_{scarred} and the M_e is the scarring probability χ , see Eq. (S6) and (S7) for details. We have added a new Fig. S3(b), reported in Fig. R1 for the Referee’s convenience, in which we plot χ versus the system size N . This shows that, for the considered system sizes, χ is rather large ($\sim 40\%$ for $N = 20$) and roughly constant with N (or at least not clearly and strongly decaying), in agreement with the observation from the Referee.

We emphasize that the M_{scarred} plotted in the inset Fig. 3(b) is computed among the 10% mid-spectrum eigenstates of the 0^+ symmetry sector only. Outside of the 10% window, and in the other special symmetry sectors (π^+ , $+\frac{\pi}{2}$, and $-\frac{\pi}{2}$), there are further scarred eigenstates. This is why the plotted M_{scarred} represents an underestimate, a point we comment on in the paragraph after Eq. S7. Beyond these technical details, the fact that $\chi \sim 40\%$ convincingly supports the statement that “the number of scarred eigenstates appears exponentially large in system size”.

FIG. R1. **Extract from Fig. S3.** [...] (b) Scarring probability χ . While the number of accessible system sizes is limited, χ is shown to be rather large, around 40% for $N = 16$ and $N = 20$, and roughly constant (or at least not strongly decaying with N), consistently with the observed exponential scaling of M_{scarred} in the inset of Fig. 3(b). Here, we consider the same parameters as in Fig. 3(b).

From this point of view, this is then perhaps not too surprising: it just says mid-spectrum eigenstates do not behave like Haar random vectors, which we should expect anyway. After all, the comparison to Haar random vectors should only be reserved for coarse grained features like expectation values of local observables and their fluctuations, one cannot expect equality to Haar random vectors at all levels of features e.g. bit-string overlaps (I find the insistence of mid-spectrum eigenstates behaving like Haar random a loose, careless argument at best, and a strawman argument at worst).

We agree that one could expect mid-spectrum eigenstates to differ from Haar random states in some way. But in what way? This is what we answer, and is highly nontrivial. Indeed, we respectfully disagree that “one should not expect equality [...] for bit-string overlaps”. That the distribution of mid-spectrum eigenstates should be Porter-Thomas, like for Haar random states, is a paradigm in quantum chaos. This expectation is corroborated in the papers mentioned in our reply above. The comparison to Haar random states is also at the essence of the phenomenon of “deep thermalization” [*Quantum* **6**, 886 (2022)]. Our work shows that scarring breaks this paradigm on the unstable periodic orbits. The Referee themselves said above that the deviation from Porter Thomas, in contrast to Haar random states, is interesting. We agree.

We thank the Referee for their valuable and constructive criticism and insights. In light of our revisions and replies, we hope that they can now recommend our paper for publication in *Nature Communications*.

Reviewers 3 and 4

I have read the authors' reply and the revised manuscript. The authors have answered many of my questions satisfactorily with extensive revisions. While they are good and the manuscript has indeed improved, I have some additional questions and comments on the quantification of genuine quantum many-body scars, which were asked by both of the referees.

We thank the Referee for once again taking the time to carefully review our paper and replies. We are glad that the Referee found our replies to many of their questions satisfactory and that the manuscript has improved. We are happy to address the further replies and comments on a point-by-point basis, below.

First, in Fig. 3(b), the authors demonstrate that the distribution of scarness for the chaotic system has a fatter tail than that for the Haar random states. However, I feel that the two distributions, while being quantitatively different, look qualitatively similar. Therefore, I would like to ask whether there are some qualitative distinctions between these two distributions. More concretely, do the tails obey different scalings with respect to scarness S or system size N ?

We thank the Referee for this question. The scaling of the tail of the histogram with N is very hard to assess. The reason is that the histogram is done over the 10% mid-spectrum eigenstates, the number of which increases with N . Only for $N = 20$ are there enough mid-spectrum eigenstates to yield a meaningful histogram (note: for smaller $N = 12$ and 16 we can nonetheless extract a meaningful number of scarred eigenstates M_{scarred} , in the inset). We shall thus focus on $N = 20$ and investigate the scaling with S . Plotting Fig. 3b in semilogarithmic scale, as done in a new Fig. S3(a) and reported in Fig. R2, suggests that both distributions have an exponential tail $\sim e^{-S/S_0}$. Fitting it, we find $S_0 = 0.83$ for the Haar random states, and $S_0 = 2.61$ for the eigenstates. We should emphasize though: the fact that the distributions appear to both have an exponential tail is by no means a limitation – the crucial point is that, due to scarring, the tail for the eigenstates decays more slowly (S_0 is more than three times larger than for the Haar random states).

Second, I still think claiming that the projection Q has a structure reflecting the underlying classical orbits for most eigenstates requires more quantitative evidence, because the authors only speculate it from 10 consecutive eigenstates.

We thank the Referee for this remark. Quantitatively, we can prove that $\approx 40\%$ of the eigenstates are scarred for $N = 16, 20$, as we show more clearly with a new Fig. S3(b) (reproduced in Fig. R2). The claim that most (i.e., more than 50%) eigenstates reflect the classical orbits is qualitative in nature and, correspondingly, so is the analysis to support it. We agree that 10 eigenstates are not enough for such analysis and we have thus added one panel to Fig. S1 (reproduced in Fig. R3), showing 50 consecutive mid-spectrum eigenstates and confirming our claim.

Other comments/questions are given below:

1. In deriving (2) from (1), do you assume $J_{\alpha\beta} = J_{\beta\alpha}$?

FIG. R2. **Extract from Fig. S3.** (a) We reproduce Fig. 3(b) from the main text for the probability distribution of the scarness S , but with logarithmic ordinate axis. This highlights an exponential scaling at large S , $\sim e^{S/S_0}$. From fits (dotted) we extract $S_0 = 0.83$ for the Haar random states, and $S_0 = 2.61$ for the eigenstates. (b) Scarring probability χ . While the number of accessible system sizes is limited, χ is shown to be rather large, around 40% for $N = 16$ and $N = 20$ [...].

FIG. R3. **Extract from Fig. S1(c).** Projection of 50 midspectrum eigenstates for the Ising model.

Yes, thanks for pointing this out. To allow $J_{\alpha\beta} \neq J_{\beta\alpha}$, the dynamics should be modified as $\frac{d\mathbf{s}_j}{dt} = [\boldsymbol{\mu} + \mathbf{J}^T \mathbf{s}_{j-1} + \mathbf{J} \mathbf{s}_{j+1}] \times \mathbf{s}_j$, with \mathbf{J}^T the transpose of \mathbf{J} . We have clarified in the text:

$$[\dots] \hat{\mathbf{s}}_j \mathbf{J} \hat{\mathbf{s}}_{j+1} = \sum_{\alpha,\beta=x,y,z} \hat{s}_j^\alpha J_{\alpha\beta} \hat{s}_{j+1}^\beta, \quad J_{\alpha\beta} = J_{\beta\alpha}, \quad \boldsymbol{\mu} \cdot \hat{\mathbf{s}}_j = \sum_{\alpha=x,y,z} \mu_\alpha \hat{s}_j^\alpha [\dots]$$

“This Hamiltonian is very general, e.g., it describes *any* homogeneous spin 1/2 chain with nearest-neighbor **reciprocal** interactions [...]”

2. The caption of Fig. 3: “In (a) and (b) the eigenstates are sampled from” \rightarrow “In (a) and (b) the eigenstates are **uniformly** sampled from”?

Correct. We have revised the text adding “**uniformly**” as suggested.

3. The authors state that the eigenstates in Fig. 3 are obtained by sampling. Does that mean you also evaluate N_{scarred} scarred from the sampling, assuming (S7)?

No, we do not sample the eigenstates to evaluate M_{scarred} (note the new notations, $N_{\dots} \rightarrow M_{\dots}$, see below). What we do is described right before Eq. S(6): we consider the M_e mid-spectrum eigenstates, say M'_e the

number of them that have S larger than the 90-th percentile of the S for the Haar random states, and define the scarring probability as $\chi = \frac{M'_e}{M_e}$. That is, we brute force consider all of the M_e mid-spectrum eigenstates, one by one, and count for M'_e . But note: considering all of the mid-spectrum eigenstates is equivalent to sampling them uniformly, which is why the inset for M_{scarred} is consistent with the rest of the figure.

4. The caption of Fig. 4: $|IS, \mathbf{y}\rangle$ and $|IS, \boldsymbol{\mu}\rangle$ do not seem to be defined.

We have clarified:

“In the top row, the system is initialized in $|IS, \mathbf{y}\rangle$, namely, on the IS UPO aligned along \mathbf{y} ($\phi = \theta = \pi/2$, marked by a star), and is more likely to be found on the same UPO at long times. In the bottom row, the system is initialized in $|IS, \boldsymbol{\mu}\rangle$, namely, on the IS UPO aligned along $\boldsymbol{\mu}$ [...]”

5. In Fig.4(b) and (c), please write the precise meaning of, e.g., $\mathbf{y} \longleftrightarrow \boldsymbol{\mu}$ and $\mathbf{y} \longleftrightarrow \mathbf{y}/\mathbf{y} \longleftrightarrow \boldsymbol{\mu}$. I think the authors should explain this in more detail.

Thanks for the suggestion, that we have implemented:

“(b,c) Scaling of \bar{Q} with system size N , with focus on the time-averaged return probabilities ($\bar{Q}(\mathbf{y})$ for $|\psi_0\rangle = |IS, \mathbf{y}\rangle$, denoted $\mathbf{y} \rightarrow \mathbf{y}$, and $\bar{Q}(\boldsymbol{\mu})$ for $|\psi_0\rangle = |IS, \boldsymbol{\mu}\rangle$, denoted $\boldsymbol{\mu} \rightarrow \boldsymbol{\mu}$) and cross probability ($\bar{Q}(\mathbf{y})$ for $|\psi_0\rangle = |IS, \boldsymbol{\mu}\rangle$, and viceversa, denoted $\mathbf{y} \longleftrightarrow \boldsymbol{\mu}$).”

6. In (S5), you use “UPO” to describe the max operation in defining scariness. Since “IS” has been used in the main text, we think that you can fix the notation.

7. The notation such as N_{scarred} , N_e , N'_e are confusing with N , so the authors could change the notations.

8. Since \hat{U} usually represents the unitary operator, I think it would be nice if the authors change the notation of the projection operator to a symmetry sector.

We thank the Referee for these suggestions on notation. We have implemented all of them, using $\max_{IS \text{ orbits}} \int Q$, M_{scarred} , M_e , M'_e , and $\hat{\mathcal{P}}$.

We thank the Referee once more for their valuable feedback and suggestions. We have replied to their questions and implemented further changes to address them. We hope that the Referee will now be able to recommend the paper for publication in *Nature Communications*.

List of changes

- Added “and the eigenstate thermalization fulfilled” to the abstract;
- In the caption of Fig. 3, we changed “is large and appears to grow exponentially with system size” into “appears exponentially large in system size”;
- Added the following passage at the end of page 2: “Being increasingly accessible in quantum computers and simulators, where for $s = 1/2$ they are often called bitstring probabilities, such projections are quickly emerging as a key object of investigation in many-body quantum chaos [48-54].”;
- Changed a few words following the suggestions by Reviewer 3 (see above for details);
- Added Fig. S1(c) to show the projection of 50 consecutive eigenstates;
- Added Fig. S3 to show further details on the scariness distribution and probability.

Reply to the Referees – 3rd Round

Reviewers 3 and 4

I have read the manuscript and reply. The authors answered our previous questions mostly satisfactorily. As the authors reply to Referees 1 and 2, whether one can find beyond-random-matrix features in the many-body wavefunction, in spite of ETH being fulfilled, is an important question. This question has a long history in terms of semiclassical chaotic systems, and the term “scar” is originally defined for such cases. We believe that this manuscript challenges the more nontrivial case for many-body systems without semiclassical limits and achieves some clear evidence for the existence of scars.

We are glad for the Referee’s positive assessment on our previous replies and on our manuscript.

That being said, we have one point that the authors should address, concerning the claim that the projection Q has a structure reflecting the underlying classical orbits for most eigenstates. In the reply, the authors state that “The claim that most (i.e., more than 50%) eigenstates reflect the classical orbits is qualitative in nature and, correspondingly, so is the analysis to support it.” However, the term “most eigenstates” sounds like almost 100% (say, asymptotically in the thermodynamic limit), which is misleading. We thus think that the authors should avoid the expression like “most eigenstates.”

The Referee is right. The word “most” can be confusing, as it can sound both as “more than 50%” or as “almost 100%”. For clarity, we have replaced “most eigenstates” with “the majority of the eigenstates”.

Therefore, after the authors address the above comment and minor comments below, we would like to recommend the manuscript for publication in Nature Communications.

We have straightforwardly addressed all the new points, and the manuscript is thus ready for publication in *Nature Communications*. We thank once again the Referee for their reviewing work and very thorough and constructive feedback and criticism, that helped us significantly strengthen the paper.

1. *The caption of Fig. 4: “In the top row, the system is initialized in $|\text{IS}, y\rangle$ ” \rightarrow “In the top row, the system is initialized in $|\text{IS}, y\rangle$ ”.*

Thanks for spotting this formatting typo. It is no longer relevant, as we have switched to a simpler $|\text{IS}_y\rangle$.

2. *We have some comments on classical chaos. In [A.S. de Wijn et al., Phys. Rev. Lett. 109, 034101 (2012)], the authors found that classical XYZ models without magnetic fields are typically chaotic. This suggests that the model considered in this manuscript is expected to be actually chaotic. In [R. Steinigeweg et al., Math. Phys. Anal. Geom. 12 (1), 19 (2009)], the authors discussed that the XXX model with $N = 4$ does not show chaos. This fact seems consistent with the fact that the Lyapunov exponent from (S37) becomes zero for $N = 4$. If these are the case, we think that the authors mention these references to strengthen their claims about the chaotic counterpart.*

Thanks for suggesting these connections, that we follow have added to our work:

“The nonlinear dynamics in Eq. (2) is generally chaotic and aperiodic [A.S. de Wijn et al., Phys. Rev. Lett. 109, 034101 (2012)]”, right after Eq. (2);

“Note as well that for $N = 4$ all the Lyapunov exponents vanish, $\lambda_{k,m} = 0$, consistently with Ref. [R. Steinigeweg et al., Math. Phys. Anal. Geom. 12 (1), 19 (2009)].”, right after Eq. S(37).

In this paper, the authors discuss the existence of quantum scars in many-body systems. Unlike the so-called "quantum many-body scars," which are atypical energy eigenstates that break eigenstate thermalization hypothesis, their quantum scars are rooted in unstable periodic orbits (UPOs) in a certain classical limit and regarded as a small correction to typical thermal eigenstates. To demonstrate this, they consider general spin- s many-body systems of N spins with nearest-neighbor interactions. For a classical limit, they consider $s \rightarrow \infty$ and obtain the corresponding classical nonlinear equation, which generally exhibits chaos and possesses unstable periodic orbits. Then, they demonstrate that such UPOs lead to quantum scars for fully quantum case ($s = 1/2$). In particular, by calculating the projection of the wave function onto the classical phase space, they find that the thermal energy eigenstates are scarred, i.e., their distributions have patterns reflecting the shape of the UPOs of the corresponding classical limits. Then, they discuss how the scars affect the long-time dynamics of isolated quantum systems, mentioning the distinctions from the classical counterpart. Finally, they conclude the paper with addressing the relevance to experiments.

In my opinion, the topic of the manuscript is interesting and worth consideration of Nature Communications. Indeed, in contrast with previous studies of quantum many-body scars, the scars considered here are in line with the original motivation of scars discussed in semiclassical systems such as quantum billiards [11]. While spin 1/2 systems do not have a well-defined classical limit, the paper shows that the classical models with $s \rightarrow \infty$ are relevant even for such systems, which I feel is surprising. While quantum scars found in this manuscript have less effects to thermalization dynamics, they provide an important clue to understand quantum many-body chaos and quantum-classical correspondence. Also, I think that the manuscript is clearly written and easy to read for non-specialists.

That being said, I hesitate to recommend the manuscript for publication in the current form since I have several concerns. First, I feel that some statements do not seem sufficiently supported, which should be modified. Second, the novelty compared with previous literature needs to be clarified more. Below, we raise questions and comments, including those related with the above concerns. Only if the authors address all of them sufficiently, I would recommend the manuscript for publication in Nature Communications.

1. The authors discuss that exponentially many of the energy eigenstates are scarred. They also mention that most of the eigenstates are scarred.
 - (a) While they are mentioned in the manuscript, no clue is discussed in the main text. The authors could briefly explain how they confirm it there.
 - (b) The above claim is discussed in Supplementary Information. There, they count numbers of eigenstates that have larger S [Eq. (S3)] compared with the average S . However, I am not sure if this criterion is appropriate. Even for random wavefunctions, the definition leads to $\chi = 50\%$, which wrongly indicates that exponentially large number of the random wavefunctions are scarred. While χ for energy eigenstates are much larger than 50%, I think the threshold of S to determine the scarred states should be much larger.
 - (c) At least, the statement that most of the eigenstates are scarred seems to be incorrect, since it would indicate $\chi \simeq 100\%$ in the thermodynamic limit.
2. The authors state that they demonstrate the ubiquity of quantum scarring.
 - (a) To what extent is it a universal phenomenon? As far as I understand, the authors only consider spin 1/2 system for quantum models (it should be clarified in the main text). What happens if they consider spin 1 systems in the form of (1)? Do they have similar scars?
For spin 1 case, we can include different type of interactions in addition to (1), such as the biquadratic interaction. I guess that this interaction considerably affects the presence of the scars since equation (2) may not hold.
 - (b) If we consider integrable spin 1/2 systems, such as Heisenberg chain, the scar will not arise, whereas $s \rightarrow \infty$ limit leads to the chaotic behavior. Is that correct?
3. Several related literature exists, including the works cited in the manuscript.
 - (a) I think that the authors should detail the distinctions and novelty compared with [15,27,30] more. In particular, Ref. [15], done by the same authors, already claims that "Here, we find the first quantum many-body scars originating from UPOs of a chaotic phase space." and "Remarkably,

these states verify the eigenstate thermalization hypothesis, and we thus refer to them as thermal quantum many-body scars." Then, what is the distinctive feature in the present manuscript compared to it?

(b) A similar motivation to connect quantum many-body chaos and unstable periodic orbits is found in, e.g., Refs. [Phys. Rev. Lett. 130, 250402 (2023); Phys. Rev. Lett. 118, 164101 (2017)]. Could you mention the relations with these works?

(c) In the famous paper by Srednicki [Phys. Rev. E 50, 888 (1994)], he also discussed the effect of genuine scars in considering quantum thermalization. I think the authors could mention this.

4. I have several questions about Q .

(a) Some additional explanations may be needed about how Q is justified as the signal of quantum scars. As mentioned by the authors, they only consider projections along UPOs in the two-dimensional manifold, although the dimension of the entire phase space is much larger (increasing with N). Is there an explanation of why this projection does not cause a problem in discussing scars? For example, for original scars in quantum billiards, such projections with large dimensional reductions are not usually employed. Moreover, in the paper by Heller (1984), the inner product between the eigenfunction and gaussian wave packets are considered, while this paper discusses the overlap with the classical states whose spins are maximally polarized. Do you have comments on these differences?

(b) While the authors mention that the high dimensionality of the classical phase space makes visualizing Q generally complicated, I think that it is not easy to compute, either, since we need to sample exponentially large numbers of points to obtain Q for many-body systems.

(c) The scaling of Q is conjectured to scale as $Ne^{-O(N)}$. I think that this scaling is important to understand the consistency with thermalization. However, no clue seems to be found for this statement.

(d) Furthermore, I feel that it is more direct to consider the scaling with s if one wants to compare with quantum scars in semiclassical systems. Namely, the role of \hbar in semiclassical systems is played by $1/s$ in the present manuscript, rather than $1/N$. So, I wonder what is the s dependence of Q .

(e) In Supplementary Information IV), they consider $s = 1/2$ in Eq. (S36) even for classical systems. How is this choice justified?

5. The authors stress that the scars are found in many-body systems.

(a) Is there any unique feature of quantum scars in many-body systems, compared with those in few-body systems (such as quantum billiards)?

(b) In the introduction, the authors state that their quantum scars yield nontrivial effects in quantum many-body systems, mentioning some examples, such as integrability, many-body localization, etc. However, this comparison looks a bit peculiar because those examples all break the eigenstate thermalization hypothesis, whereas quantum scars presented in this manuscript do not.

6. The authors claim that their predictions are straightforwardly verified in experiments. However, as mentioned in the manuscript, the effect decays exponentially as N is increased. Therefore, it would be hard to detect as we consider many-body limit. The authors should comment on this apparent problem.

7. In calculating (S3), is it easy to take the maximum over all UPOs?

8. After (S32), they find $6N$ eigenvalues, although the original problem (S21) will have $3N$ eigenvalues. How do they reconcile with each other?

9. In (S34), λ does not depend on N . Is it because the authors consider perturbations around specific UPOs (i.e., IS UPOs)?

10. There are some typos.

(a) On page 2, "scarring can be expected when $\lambda\omega < 1$ " \rightarrow "scarring can be expected when $\lambda/\omega < 1$ "

(b) In (S1) and (S2), some $\sigma_j^z \sigma_{j+1}^z$ should be $\sigma_j^x \sigma_{j+1}^x$ or $\sigma_j^y \sigma_{j+1}^y$.

(c) In Fig. S1, 4181 and 4182 would be 4182 and 4183, respectively.

I have read the authors' reply and the revised manuscript. The authors have answered many of my questions satisfactorily with extensive revisions. While they are good and the manuscript has indeed improved, I have some additional questions and comments on the quantification of genuine quantum many-body scars, which were asked by both of the referees.

First, in Fig. 3(b), the authors demonstrate that the distribution of scariness for the chaotic system has a fatter tail than that for the Haar random states. However, I feel that the two distributions, while being quantitatively different, look qualitatively similar. Therefore, I would like to ask whether there are some qualitative distinctions between these two distributions. More concretely, do the tails obey different scalings with respect to scariness S or system size N ?

Second, I still think claiming that the projection Q has a structure reflecting the underlying classical orbits for most eigenstates requires more quantitative evidence, because the authors only speculate it from 10 consecutive eigenstates.

Other comments/questions are given below:

1. In deriving (2) from (1), do you assume $J_{\alpha\beta} = J_{\beta\alpha}$?
2. The caption of Fig. 3: "In (a) and (b) the eigenstates are sampled from" \rightarrow "In (a) and (b) the eigenstates are uniformly sampled from"?
3. The authors state that the eigenstates in Fig. 3 are obtained by sampling. Does that mean you also evaluate N_{scarred} from the sampling, assuming (S7)?
4. The caption of Fig. 4: $|\text{IS}, y\rangle$ and $|\text{IS}, \mu\rangle$ do not seem to be defined.
5. In Fig.4 (b) and (c), please write the precise meaning of, e.g., $y \leftrightarrow \mu$ and $(y \rightarrow y)/(y \leftrightarrow \mu)$. I think the authors should explain this in more detail.
6. In (S5), you use "UPO" to describe the max operation in defining scariness. Since "IS" has been used in the main text, we think that you can fix the notation.
7. The notation such as $N_{\text{scarred}}, N_e, N'_e$ are confusing with N , so the authors could change the notations.
8. Since \hat{U} usually represents the unitary operator, I think it would be nice if the authors change the notation of the projection operator to a symmetry sector.

I have read the manuscript and reply. The authors answered our previous questions mostly satisfactorily. As the authors reply to Referees 1 and 2, whether one can find beyond-random-matrix features in the many-body wavefunction, in spite of ETH being fulfilled, is an important question. This question has a long history in terms of semiclassical chaotic systems, and the term "scar" is originally defined for such cases. We believe that this manuscript challenges the more nontrivial case for many-body systems without semiclassical limits and achieves some clear evidence for the existence of scars.

That being said, we have one point that the authors should address, concerning the claim that the projection Q has a structure reflecting the underlying classical orbits for most eigenstates. In the reply, the authors state that "The claim that most (i.e., more than 50%) eigenstates reflect the classical orbits is qualitative in nature and, correspondingly, so is the analysis to support it." However, the term "most eigenstates" sounds like almost 100% (say, asymptotically in the thermodynamic limit), which is misleading. We thus think that the authors should avoid the expression like "most eigenstates."

Therefore, after the authors address the above comment and minor comments below, we would like to recommend the manuscript for publication in Nature Communications.

1. The caption of Fig. 4: "In the top row, the system is initialized in $|IS, y\rangle$ " \rightarrow "In the top row, the system is initialized in $|IS, y\rangle$ "
2. We have some comments on classical chaos. In [A.S. de Wijn et al., Phys. Rev. Lett. 109, 034101 (2012)], the authors found that classical XYZ models without magnetic fields are typically chaotic. This suggests that the model considered in this manuscript is expected to be actually chaotic. In [R. Steinigeweg et al., Math. Phys. Anal. Geom. 12 (1), 19 (2009)], the authors discussed that the XXX model with $N = 4$ does not show chaos. This fact seems consistent with the fact that the Lyapunov exponent from (S37) becomes zero for $N = 4$. If these are the case, we think that the authors mention these references to strengthen their claims about the chaotic counterpart.